# Dimensional Collapse in Transformer Attention Outputs: A Challenge for Sparse Dictionary Learning

**Junxuan Wang** [1 2]  **Xuyang Ge** [1 2]  **Wentao Shu** [1 2]  **Zhengfu He** [1 2]  **Xipeng Qiu** [1 2]

## Abstract

Transformer architectures, and their attention mechanisms in particular, form the foundation of modern large language models. While transformer models are widely believed to operate in high-dimensional hidden spaces, we show that attention outputs are confined to a surprisingly low-dimensional subspace, with an effective dimensionality of only about 60% of the full space—a phenomenon that is consistently observed across diverse model families and datasets, and arises from overlap among the output subspaces of different attention heads. Critically, we find this low-rank structure as a key factor of the prevalent dead feature problem in sparse dictionary learning, where it creates a mismatch between randomly initialized features and the intrinsic geometry of the activation space. Building on this insight, we propose a subspace-constrained training method for sparse autoencoders (SAEs), initializing feature directions into the active subspace of activations. Our approach reduces dead features from 87% to below 1% in Attention Output SAEs with 1M features, and can further extend to other sparse dictionary learning methods. Our findings provide both new insights into the geometry of attention and practical tools for improving sparse dictionary learning in large language models. Code is available at https://github.com/OpenMOSS/Llamascopium.

## 1. Introduction

Over the past years, mechanistic interpretability has shifted from a collection of proof-of-concept tools (Olsson et al., 2022; Wang et al., 2022; Meng et al., 2023; Gould et al.,

2023) toward a fast-growing, scale-driven field (Ameisen et al., 2025; Lindsey et al., 2025). This transformation is driven by a wave of sparse dictionary learning methods–such as sparse autoencoders (SAEs) and their variants (Cunningham et al., 2023; Bricken et al., 2023b; Lindsey et al., 2024b), transcoders (Dunefsky et al., 2024; Ge et al., 2024), and low-rank sparse attention (He et al., 2025)–that once targeted small models but are now being pushed to larger architectures and wider model families (Templeton et al., 2024; Gao et al., 2024; Hazra et al., 2025). As these approaches scale in performance and model coverage, they provide increasingly complete and fine-grained explanations of neural network behavior (Lindsey et al., 2024a; Gao et al., 2024).

However, scaling these approaches presents substantial practical challenges (Templeton et al., 2024; Gao et al., 2024; Mudide et al., 2025). As models and feature dictionaries grow, the parameter count increases rapidly, leading to significant computational overhead. Moreover, a large fraction of learned features remain inactive, resulting in considerable waste in both computation and memory (Templeton et al., 2024; Kissane et al., 2024). In this work, we identify **the low-rank structure of the activations as a primary driver of dead features** (Section 5.1).

Through singular value decomposition and effective dimensionality analyses (Roy & Vetterli, 2007; Staats et al., 2025), we show for the first time that **the outputs of multi-head self-attention in transformer-based language models exhibit a remarkably low-rank structure** (Section 4). Compared to multilayer perceptron (MLP) outputs and residual streams, attention outputs consistently concentrate in a low-dimensional subspace. We show that this behavior is robust across layers, datasets, and model families, including GPT-2 (Radford et al., 2019), Llama 3.1 (Dubey et al., 2024), Gemma 2 (Rivière et al., 2024), and Qwen 3 (Yang et al., 2025). This universality aligns with prior observations of shared structures across models (Olah et al., 2020; Chughtai et al., 2023; Gurnee et al., 2024; Wang et al., 2025), while revealing a distinct form of representation collapse (Hua et al., 2021; Jing et al., 2022) at the level of attention outputs. We trace this low-rank structure to the anisotropy of the output projection matrix $W^O$ and the overlap among

---

[1]Shanghai Innovation Institute, Shanghai, China [2]OpenMOSS Team, School of Computer Science, Fudan University, Shanghai, China. Correspondence to: Xipeng Qiu <xpqiu@fudan.edu.cn>.

*Proceedings of the 43rd International Conference on Machine Learning*, Seoul, South Korea. PMLR 306, 2026. Copyright 2026 by the author(s).

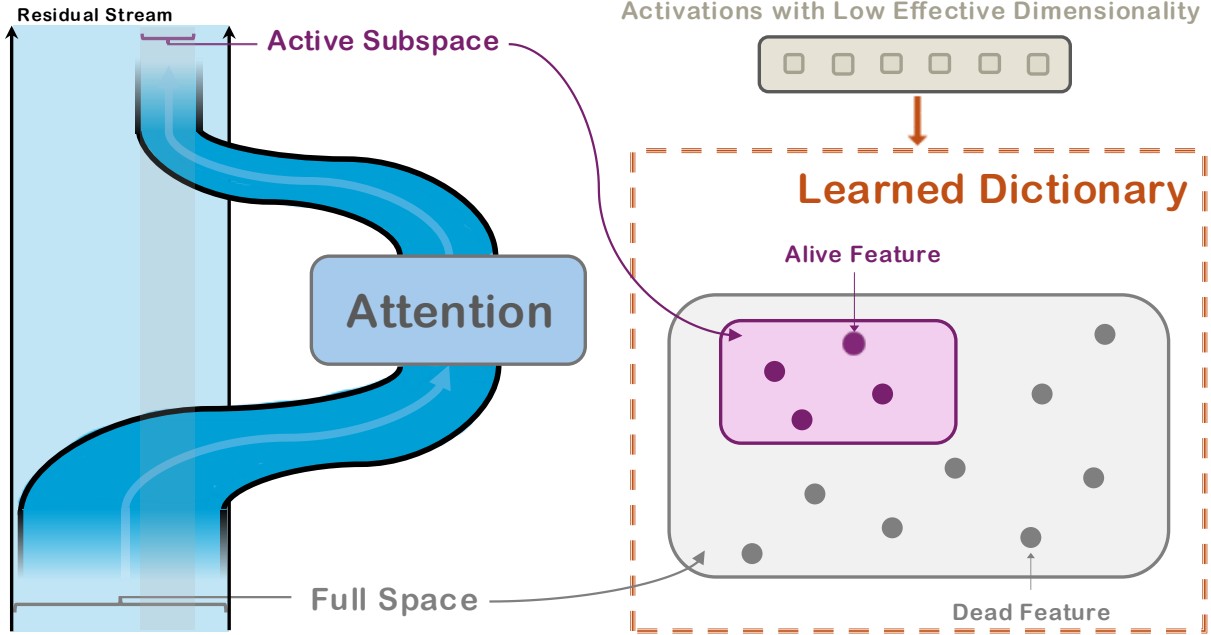

*Figure 1.* (left) Attention outputs exhibit pronounced low-rank structure compared to residual streams and MLP outputs, indicating that the attention layer writes into a subspace of the residual stream. (right) Low effective dimensionality of activations is a root cause of dead features in sparse dictionary learning methods. Setting feature directions in the active subspace mitigates this issue.

output subspaces of different attention heads.

In Section 5, we investigate how the low-rank structure of attention outputs interacts with SAE training. By evaluating the full suite of open-source SAEs from *LlamaScope* (He et al., 2024), we show that low effective dimensionality strongly correlates with the number of dead features, suggesting a mismatch between random initialization and the low-dimensional geometry of the activations. Drawing inspiration from Phan et al. (2025)'s principal component initialization for the first network layer, we propose *Active Subspace Initialization*, which aligns SAE features with the active subspace of activations, **substantially reducing dead features while improving reconstruction**. Following Lindsey et al. (2024a) and Gao et al. (2024), we conduct scaling experiments, which further reveal that ASI achieves superior reconstruction across feature counts, and when combined with SparseAdam[1], it achieves the best reconstruction in large scale and reduces dead features from 87% to below 1% in Attention Output SAEs with 1M features trained on Llama-3.1-8B (Dubey et al., 2024).

Furthermore, we show that **Active Subspace Init can generalize to sparse replacement models** (He et al., 2025; Dunefsky et al., 2024; Ameisen et al., 2025) (Section 5.4). When applied to other sparse dictionary learning methods, our initialization procedure systematically reduces the prevalence of dead parameters across architectures.

---

[1]See the SparseAdam documentation for details.

## 2. Related Work

### 2.1. Representation Collapse in Neural Representations and Low-Rankness in Attention Mechanism

A long line of research has shown that neural representations frequently concentrate in low-dimensional subspaces, forming representation collapse (Hua et al., 2021; Tian et al., 2021; Jing et al., 2022). These works generally focus on visual models trained using self-supervised methods.

Within attention mechanisms, prior work has investigated various notions of "low-rankness": low-rank approximation of attention patterns (Wang et al., 2020; Tay et al., 2020; Raganato et al., 2020), low-rank parameterization for model compression (Noach & Goldberg, 2020; Hu et al., 2022), and the low-rank bottleneck in single-head outputs (Bhojanapalli et al., 2020; Qiu et al., 2025).

Different from these prior lines of work, we demonstrate that the multi-head self-attention outputs exhibit a low-rank structure, revealing a distinct and under-explored phenomenon.

### 2.2. Superposition Hypothesis and Sparse Dictionary Learning Methods

The superposition hypothesis posits that neurons encode multiple non-orthogonal underlying features (Arora et al., 2018; Olah et al., 2020; Elhage et al., 2022; Park et al., 2024). Motivated by this view, a variety of sparse dictionary learn-

ing methods have been developed for interpretability, including sparse autoencoders and their variants (Cunningham et al., 2023; Bricken et al., 2023b; Lindsey et al., 2024b), transcoders (Dunefsky et al., 2024; Ge et al., 2024), and low-rank sparse attention (He et al., 2025). These approaches decompose activations into sparse combinations of learned features while differing in their mechanisms for predicting or approximating feature activations. Their successful application across a wide range of model scales (Templeton et al., 2024; Lieberum et al., 2024; He et al., 2024), architectures (Wang et al., 2025), and modalities (Abdulaal et al., 2024) highlights their practical effectiveness for interpretability; however, they do not constitute direct hypothesis tests of the superposition hypothesis, which remain active topics of debate (Sharkey et al., 2025).

### 2.3. Dead Features in Sparse Dictionary Learning Methods

A persistent challenge in sparse dictionary learning methods is the emergence of *dead features*[2] (Templeton et al., 2024; Kissane et al., 2024), which are also referred to as *dead units* in sparse replacement models (Dunefsky et al., 2024; Ge et al., 2024; He et al., 2025). These features contribute nothing to reconstruction quality, wasting parameters and computation. Existing approaches to mitigate this issue rely on auxiliary loss terms (Gao et al., 2024; Conerly et al., 2025) or resampling strategies (Bricken et al., 2023b) to encourage feature usage.

### 2.4. PCA-Inspired Network Initialization

A common practice applies PCA to input data for dimensionality reduction before network training (Hastie et al., 2009; Montavon et al., 2012; Jolliffe, 1986; Nasrabadi, 2007). Recently, Phan et al. (2025) proposed *PCsInit*, which initializes the first layer weights of networks with top principal components of data—embedding the PCA transform directly into the network. This provides the model with a superior parameter set (Gu et al., 2025), boosting performance by construction.

## 3. Preliminaries

### 3.1. Multi-Head Self-Attention and Notations

We consider a Transformer block with multi-head self-attention (MHSA) (Vaswani et al., 2017). Given input activations $X \in \mathbb{R}^{n \times d}$, where $n$ is the token count and $d$ is the model hidden size, each attention head $i$ computes:

$$Q_i = XW_i^Q, \quad K_i = XW_i^K, \quad V_i = XW_i^V,$$
$$W_i^Q, W_i^K, W_i^V \in \mathbb{R}^{d \times d_h},$$

---

[2]Following Bricken et al. (2023b), we define a feature as dead if it never activates over 10 million tokens in this paper.

where $d_h = d/H$ is the dimensionality of each head, and $H$ is the total number of heads. The attention weights and head outputs are then given by:

$$A_i = \text{softmax}\left(\frac{Q_i K_i^\top}{\sqrt{d_h}}\right), \quad Z_i = A_i V_i \in \mathbb{R}^{n \times d_h}.$$

Let $Z = \text{Concat}[Z_1, \dots, Z_H] \in \mathbb{R}^{n \times d}$ denote the concatenated outputs of all attention heads (Nanda & Bloom, 2022). The final **attention output** is obtained by applying the output projection:

$$O = ZW^O = [Z_1, \dots, Z_H][W_1^O, \dots, W_H^O]^\top$$
$$= \sum_{i=1}^{H} Z_i W_i^O = \sum_{i=1}^{H} O_i,$$

where each $W_i^O \in \mathbb{R}^{d_h \times d}$ is the corresponding submatrix of $W^O \in \mathbb{R}^{d \times d}$ associated with each head $i$.

This formulation makes explicit that $O$ is the sum of the outputs from all heads, where each head produces a rank-$d_h$ output that is projected into the residual stream space through its corresponding $W_h^O$. Thus, $O$ represents the attention block's total contribution to the residual stream.

### 3.2. TopK Sparse Autoencoders

In this work, we adopt the TopK sparse autoencoder (TopK SAE) introduced by Gao et al. (2024). Unlike standard SAEs that impose an $\ell_1$ penalty, TopK SAE enforces exact sparsity by keeping only the top-$k$ activations in the latent representation for each input. Formally, given an input vector $x \in \mathbb{R}^d$, the encoder produces

$$z = \text{TopK}(W_e x + b_e),$$

where $\text{TopK}(v)$ sets to zero all but the largest $k$ entries of $v$. The decoder then reconstructs

$$\hat{x} = W_d z + b_d.$$

The model is trained to minimize the reconstruction loss, optionally augmented with an auxiliary loss to prevent dead latents:

$$\mathcal{L}_{\text{TopK-SAE}} = \|x - \hat{x}\|_2^2 + \alpha \cdot \mathcal{L}_{\text{aux}},$$

where $\mathcal{L}_{\text{aux}}$ is an optional term designed to penalize latents that never activate over a training period, and $\alpha$ balances reconstruction fidelity and latent utilization.

## 4. Low-Rank Structure of Attention Outputs

We begin by presenting our central empirical finding: in Transformer models, attention outputs consistently display the strongest low-rank structure compared to MLP outputs and residual streams. As shown in Figure 2, attention outputs have a significantly lower effective rank. This phenomenon is remarkably robust, holding across different

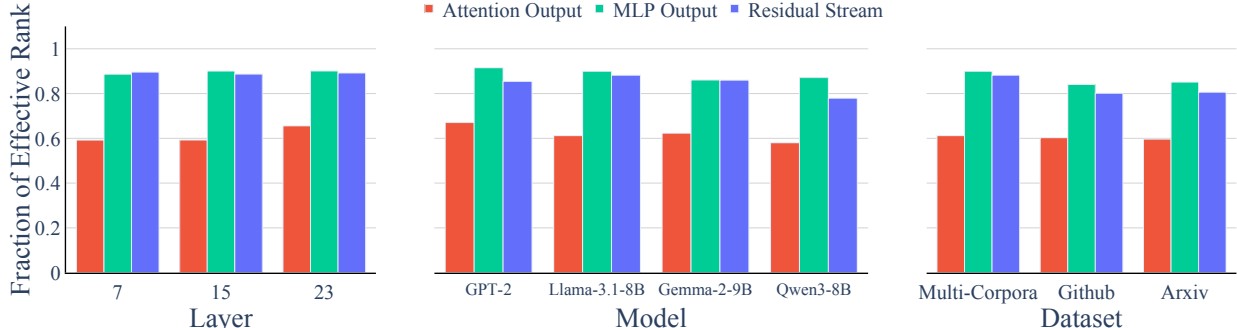

*Figure 2.* Across layers, model families and datasets, attention outputs exhibit dramatically lower effective rank than residual streams and MLP outputs, indicating that the attention layer writing into a low dimensional subspace of residual stream is a universal phenomenon. Details in Section 4.1. (left) Evaluation of Llama-3.1-8B on SlimPajama (Soboleva et al., 2023) dataset. (mid) Middle-layer analysis across model families on SlimPajama dataset. (right) Middle-layer analysis of Llama-3.1-8B across datasets.

layers, model families and datasets. Further details regarding the activation sources are provided in Section 4.2 and Appendix A. These observations highlight that the attention block modifies a subspace of the residual stream, while the MLP operates nearly on the full space.

### 4.1. Quantifying Low-Rankness with Effective Rank

We consider the activation matrix $A \in \mathbb{R}^{n \times d}$, where each row corresponds to the activation vector of a single token. Here, $n$ denotes the number of data points and $d$ the dimensionality of the activation space (e.g., the model's hidden size). Unless otherwise specified, $A$ represents mean-centered activations.

To quantify the effective dimensionality of the activations, we adopt the ***effective rank*** metric introduced by Roy & Vetterli (2007).

**Definition 4.1** (Effective Rank, Roy & Vetterli (2007))**.** Let $\widetilde{A}$ be a nonzero matrix with singular value decomposition $\widetilde{A} = U\Sigma V^\top$, where $\Sigma = \mathrm{diag}(\sigma_1, \sigma_2, \ldots, \sigma_r)$ contains singular values in descending order. Define the normalized singular value distribution as

$$p_k = \frac{\sigma_k}{\sum_{j=1}^r \sigma_j}, \quad k = 1, 2, \ldots, r.$$

The (Shannon) entropy of this distribution is

$$H(p_1, p_2, \ldots, p_r) = -\sum_{k=1}^r p_k \log p_k.$$

Then, the *effective rank* of $\widetilde{A}$ is defined as

$$\mathrm{erank}(\widetilde{A}) = \exp\{H(p_1, p_2, \ldots, p_r)\}.$$

Intuitively, the effective rank captures how evenly the singular values are distributed—higher values indicate a more isotropic spectrum, whereas lower values reflect concentration along a few dominant directions. *Fraction of effective rank* used in Figure 2 means effective rank divided by the dimension of activation space.

Following Bricken et al. (2023b) and Rajamanoharan et al. (2024a), we compute the ***fraction of downstream loss recovered*** by varying the number of retained components. We decompose the activations into singular value components and take the language model loss under full activation ablation as a baseline. As components are progressively reintroduced, we report the fraction of this ablated loss that is recovered. See Appendix B for a formal definition.

These quantitative measures complement our core analyses by providing a numerical characterization of the low-rank structure present in activations.

### 4.2. Experiment Settings

For each activation type, we collect a total of 10 million activation vectors.[3] We empirically verified this sample size suffices to ensure stable and reproducible singular spectrum analyses in Appendix C.

Unless otherwise specified, all experiments in Section 4 run on the middle layer of Llama-3.1-8B (layer 15, zero-indexed), using SlimPajama dataset.

### 4.3. Empirical Evidence of Low-Rank Structure

We draw our findings from three lines of evidence:

**Low Effective Rank of Attention Output**    Attention outputs have a effective rank of around 60% of the total dimensionality. In contrast, MLP outputs and the residual streams

---

[3]In extremely rare cases, outlier activations inflate variance along certain directions, potentially biasing variance-based dimensionality estimates. To mitigate this, we exclude activations whose norms exceed $5\sigma$ from the mean.

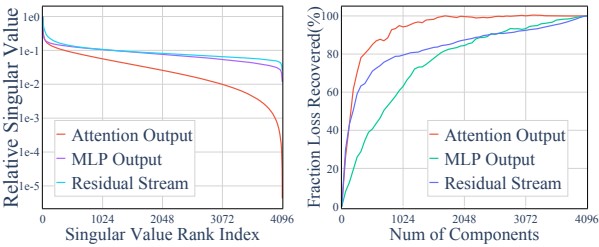

*(a)* Singular Value Spectra. *(b)* Fraction of Loss Recovered.

*Figure 3.* **(a)** The attention output is the most low-rank, as indicated by the sharpest decay in singular values. **(b)** Fraction of loss recovered using varying numbers of top singular components.

show much higher effective rank around 90% (Figure 2).

**Rapid Singular Spectral Decay in Attention Output** This is quantitatively evidenced by the number of components retaining significant energy: only 74.7% singular values exceed 1%[4] of the maximum in attention output, versus 100.0% for MLP output and residual stream (Figure 3a).

**Efficient Downstream Loss Recovery** Compared to zero ablation, attention output requires only 39.1% of dimensions to recover over 99% of the downstream loss, versus 95.3% and 96.9% of the dimensions for MLP outputs and residual streams to recover the same proportion (Figure 3b).

More results of these metrics across different layers, models, datasets and activation positions are shown in Appendix D.

### 4.4. Tracing the Origin of this Low-Rank Structure

Among all activation types, attention outputs consistently exhibits the most rapid singular spectral decay. To investigate whether this low-rank structure originates from the attention heads outputs ($Z$), the output projection matrix ($W^O$), or their interaction, we perform a decomposition-based analysis.

Recall that the attention output is computed as $O = ZW^O$, where $Z \in \mathbb{R}^{n \times d}$ is the concatenated output of attention heads, and $W^O \in \mathbb{R}^{d \times d}$ is a linear projection. To know how the singular value spectra of $O$ is shaped, we analyze the variance[5] of $O$ along a direction $\hat{e} \in \mathbb{R}^d$, given by:

$$\mathrm{Var}(O\hat{e}) = \mathrm{Var}(ZW^O\hat{e}).$$

This expression highlights that the variance along $e$ is determined by two factors: the norm of $W^O\hat{e}$ and the variance of $Z$ along $W^O\hat{e}$. Specifically, we can rewrite the variance as:

$$\mathrm{Var}(O\hat{e}) = \mathrm{Var}(Z\hat{v}) \cdot \|v\|_2^2 \text{ where } v = W^O\hat{e}, \ \hat{v} = \frac{v}{\|v\|_2}.$$

---

[4]The resolution of bfloat16 (Kalamkar et al., 2019) is 0.01.

[5]For zero-mean activations, singular values correspond to the standard deviations of activations along the associated singular directions.

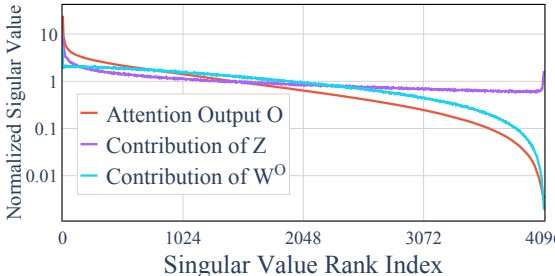

*Figure 4.* Decomposition of singular value spectra in attention output $O$. We analyze the contributions of the *concatenated head outputs Z* and the *projection matrix $W^O$* to the singular value of $O$ ($=ZW^O$). For each component, the red value is the product of the purple and blue values. The curve of $O$ closely follow that of $Z$ for the top components, whereas its downward trend at the tail is mainly due to $W^O$ contribution.

We refer to $\mathrm{Var}(Z\hat{v})$ as the **contribution of** $Z$, capturing how much variance the head output $Z$ provides in that direction, and $\|v\|_2^2$ as the **contribution of** $W^O$, measuring how much the output projection $W^O$ scales or suppresses that direction.

We compute and visualize the singular values of attention output $O$ and these two contributions in Figure 4. This analysis reveals that the low-rank structure of attention outputs is strongly influenced by $W^O$, which compresses $Z$ into a lower-dimensional subspace. The effective rank of $Z$ in Figure 11 further supports this conclusion.

From a mechanistic perspective, an intuitive explanation is that although each attention head contributes a $d_{\text{head}}$-dimensional subspace, the superposition of attention heads (Jermyn et al., 2024; He et al., 2025) inevitably induces overlaps among these subspaces. Let $O_i$ denote the output of the $i^{\text{th}}$ attention head. Consequently, the dimension of the MHSA output satisfies

$$\dim\left(\bigcup_i \mathrm{span}(O_i)\right) \leq \sum_i \dim(\mathrm{span}(O_i))$$
$$= d_{\text{head}} \cdot n_{\text{head}}$$
$$= d_{\text{model}} \quad \text{(in standard MHSA)}.$$

We further validate this interpretation by measuring the effective rank of each head output $O_i$, with detailed results reported in Table 1.

## 5. Active Subspace Initialization for Sparse Autoencoders

### 5.1. Empirical Correlation Between Low-Rank Structure and Dead Features

To study how low-rankness affects the interpretability of attention, we adopt the same framework and dataset as the

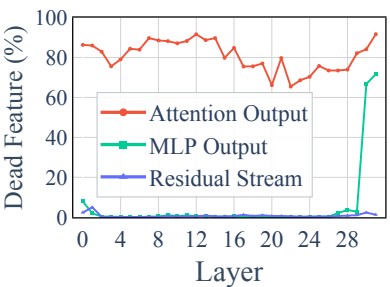 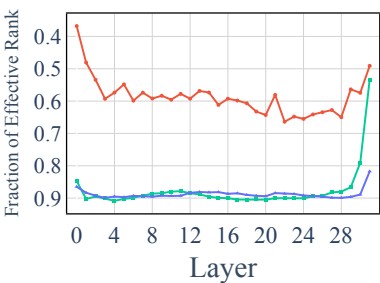 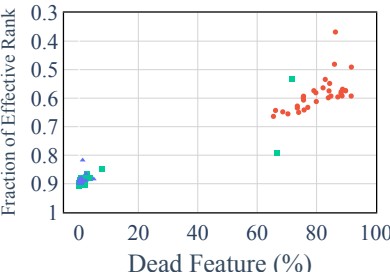

*Figure 5.* The number of dead features (left) and the effective rank (mid) of each activation in Llama-3.1-8B, shows a surprising consistency (right): activations with lower effective rank have more dead features, corresponding to all layers of attention output and last two layers of MLP output.

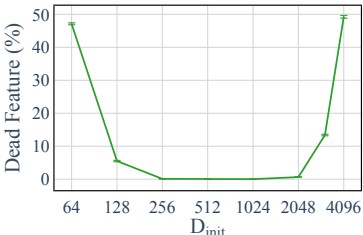 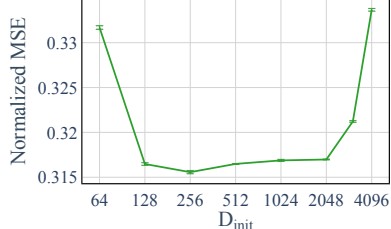 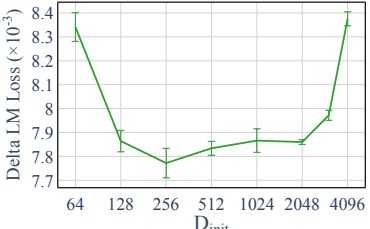

*Figure 6.* After using ASI, proportion of dead features (left), normalized MSE (mid) and Delta LM loss (right) across different $d_{init}$ for activations with a full space dimension of 4096. All experiments repeat 3 times using different random seeds and error bars indicate mean ± std.

original LlamaScope study (He et al., 2024) to evaluate their SAEs trained on attention outputs, MLP outputs, and the residual stream[6]. We find that the number of dead features is strongly related to effective rank, as shown in Figure 5. This observation suggests that dead features may stem from the low-rank geometry of the activation space. We also train SAEs using different SAE hyperparameters and further systematically verify that this phenomenon is prevalent in Appendix F.

### 5.2. Active Subspace Initialization for Sparse Autoencoders

Based on this observation, we propose *Active Subspace Initialization* (ASI), a lightweight and generalizable strategy for scaling SAEs to high capacities. Let $d$ denote the input dimension, $h$ the hidden dimension of the SAE, and $n$ the number of data points. Given activation matrices $\widetilde{A} \in \mathbb{R}^{n \times d}$ with singular value decomposition $\widetilde{A} = U\Sigma V^\top$ and $V \in \mathbb{R}^{d \times d}$ contains the right singular vectors, we select the top $d_{init}$ singular vectors to define the active subspace:

$$V_{active} = V_{:,:d_{init}} \in \mathbb{R}^{d \times d_{init}}.$$

To initialize the SAE within this subspace, we first randomly initialize the decoder weights $W_D \in \mathbb{R}^{h \times d}$ and then *project*

*their first $d_{init}$ columns onto the active subspace*:

$$W_D \leftarrow W_D \, V_{active} \, V_{active}^\top, \qquad W_E = W_D^\top.$$

where $W_E$ is the encoder weight matrix and $W_D$ is the decoder weight matrix. Intuitively, ASI aligns the initial SAE parameters with the effective directions of the data, ensuring that SAEs start in a meaningful low-dimensional subspace. As $d_{init}$ decreases from the full space dimension[7] within a certain range, the number of dead features in the Attention Output SAE rapidly drops, with a corresponding improvement in Mean Square Error (MSE) and Delta LM loss[8] (Figure 6). We refer readers to Appendix E for full SAE training details. Additional ablations, including the random subspace initialization baseline and the effect of activation rank, are reported in Appendix J. Pseudocode for Active Subspace Initialization (ASI) is provided in Appendix K.

Using **Active Subspace Initialization** offers several benefits:

---

[6]Another prominent set of open-source SAEs, GemmaScope (Lieberum et al., 2024), train their attention SAEs on Z rather than attention output.

[7]Setting $d_{init}$ equal to the full space dimension is equivalent to not using Active Subspace Initialization.

[8]Following He et al. (2024), this metric is defined as the difference between the original language model loss and the loss when the SAE is inserted at the corresponding position, evaluated over 1 million tokens.

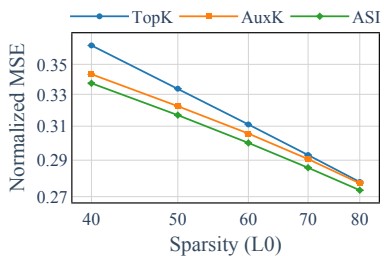
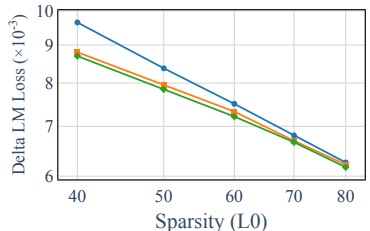

*Figure 7.* At a fixed number of features ($n = 32768$), ASI (TopK SAE with Active Subspace Init) achieves a better reconstruction-sparsity trade-off than TopK (standard TopK SAE) and AuxK (TopK SAE with auxiliary loss). A similar trend is observed in its impact on Delta LM Loss. All experiments repeat 3 times using different random seeds and show the mean. The results of std are in Appendix G due to resolution constraints.

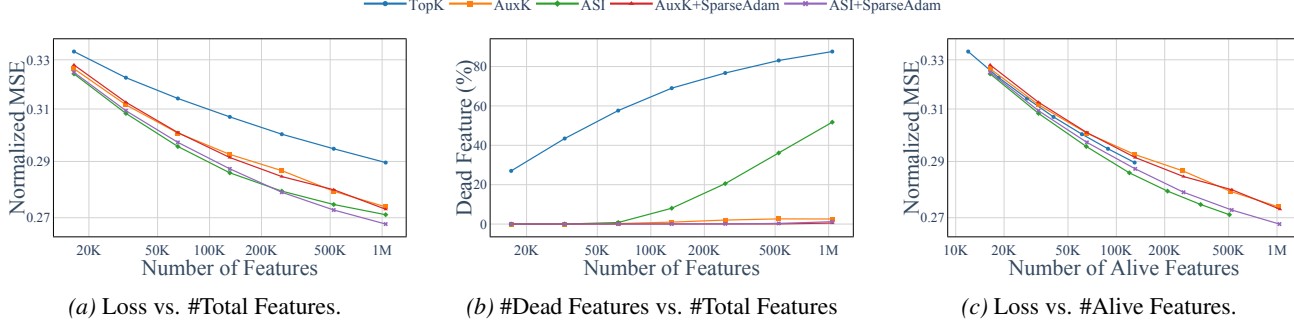

*(a) Loss vs. #Total Features.*  *(b) #Dead Features vs. #Total Features*  *(c) Loss vs. #Alive Features.*

*Figure 8.* Scaling results of TopK SAEs and their variants enhanced with *AuxK*, *Active Subspace Init*, and *SparseAdam*–all trained on attention output from the middle layer of Llama-3.1-8B. (A) Loss at convergence across different feature counts: Active Subspace Init consistently achieves lower reconstruction error than TopK and AuxK. Active Subspace Init with SparseAdam achieves the best at large scale. (B) Dead features: Active Subspace Init reduces dead features compared to TopK, but still retains many at extremely large scales. Enhanced with SparseAdam, dead features can be reduced to less than 1%. (C) Loss across different number of **alive** features: Active Subspace Init achieves the most efficient utilization of **alive** features, while AuxK shows the lowest efficiency. Details in Section 5.3.

**Reduced Dead Features and Enhanced Sparsity-Reconstruction Frontier Without Additional Compute** It achieves near-zero dead features and slightly superior results compared to the auxiliary loss approach (AuxK), at no additional computational cost of the same order. (Figure 7).

**Optimal Scaling Characteristics** Our approach demonstrates optimal scaling behavior across various SAE training methods. It outperforms TopK and AuxK in any evaluated scale, from 16K to 1M features (Section 5.3).

**General Applicability** The technique maintains applicability to diverse architectural variants and activation functions, as it operates directly on the intrinsic properties of activations. This generalizability is further explored in Section 5.4 and Appendix L.

We conduct a statistical significance test in Appendix H to demonstrate the statistical significance of our conclusions. We validate the effectiveness of ASI across different layers, models, and datasets in Appendix I. We further compare the features of TopK and ASI in Appendix N, analyzing both the degree of monosemanticity and the behavior of SAE fea-

tures in the dead subspace, to ensure that ASI increases the number of alive features **while preserving feature quality and maintaining the dictionary's coverage and reconstruction performance in the dead subspace.**

### 5.3. Scaling Laws

To assess scaling, we evaluate our method as the number of SAE features grows from 16K to 1M, keeping other hyperparameters fixed (see Appendix E).

**Active Subspace Init Improves Reconstruction.** As shown in Figure 8a, Active Subspace Init consistently outperforms TopK and AuxK across all scales.

**Caveat: Some Dead Features Remain at Extremely Large Scales in Active Subspace Init.** Figure 8b shows that, when scaling to extremely large feature counts, Active Subspace Init produces more dead features than AuxK. However, reconstruction performance remains better, indicating that the revived features from AuxK contribute little to actual reconstruction quality (Figure 8c).

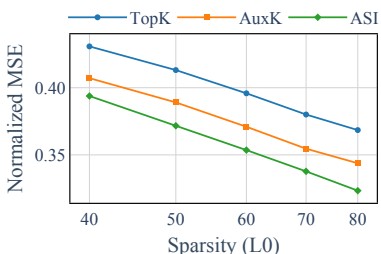 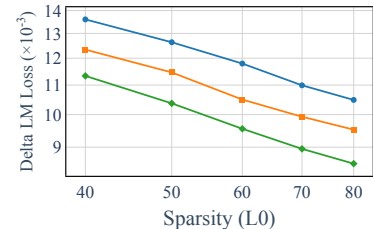

*Figure 9.* At a fixed number of Lorsa heads ($n = 32768$), ASI (TopK Lorsa with Active Subspace Init) achieves a better reconstruction-sparsity trade-off than TopK (standard TopK Lorsa) and AuxK (TopK Lorsa with auxiliary loss). A similar trend is observed in its impact on Delta LM Loss.

**Use Active Subspace Init with SparseAdam Further Improves Performance.** Prior work (Bricken et al., 2023a) identified *stale momentum* as a key factor in dead feature formation. Building on this insight, we propose using **SparseAdam**, an optimizer specifically designed for sparse activation settings. By updating only the momentum terms and parameters corresponding to non-zero gradients, SparseAdam avoids stale momentum and thus mitigates the dead feature issue. As shown in Figures 8a, 8b, combining Active Subspace Init with SparseAdam substantially reduces dead features while reaching the lowest reconstruction error. While orthogonal to our initialization method, this choice provides a practical complement that further stabilizes training when scaling SAEs to very large capacities. We discuss more about *stale momentum* and **SparseAdam** in Appendix M.

### 5.4. Generalizing to Sparse Replacement Models

Recent work by He et al. (2025) shows that Lorsa—a sparse replacement for MHSA—has a substantial fraction of dead parameters. We hypothesize this is partly due to initialization: standard random initialization ignore the low-dimensional active subspace of attention outputs (Section 4).

**Applying ASI to Lorsa.** To test whether our subspace-based approach extends beyond SAEs, we incorporate ASI (Section 5.2) into Lorsa training. When training Lorsa to approximate a target MHSA, we partition Lorsa heads into such groups, matching the number of attention heads in the MHSA. For each group, we initialize the encoder and decoder matrices directly within the corresponding input and output active subspaces of the MHSA head it corresponds. In addition, we initialize each group's Q/K weights from the MHSA Q/K parameters. Pseudo code is provided in Appendix K

**Results.** This initialization sharply reduces dead parameters under identical hyperparameters (Figure 9) while improving reconstruction quality. Further Lorsa training details are in Appendix O.

## 6. Discussion and Limitations

**Low-rank attention outputs suggest a new direction for model improvement.** Modern large language models are largely built by stacking attention and MLP blocks with residual connections. Our finding that attention outputs are low-rank suggests a new avenue for improving model capacity and expressivity: explicitly mitigating rank collapse within attention modules. This phenomenon persists across models with and without *grouped-query attention* (GQA), and under both relative and absolute positional encodings (Section 4), suggesting that it reflects a fundamental property of attention rather than an artifact of specific design choices. We also make preliminary attempts to mitigate this low-rankness, but with limited success; these negative results are documented in Appendix **??**.

**Causality between Low-Rank Structure and Dead Features.** We find a strong correlation between low-rank activations and the emergence of dead features (Section 5), but the underlying causal mechanism is unresolved. This effect may arise from optimization dynamics or feature competition, and we leave a rigorous explanation to future work.

**When to Use Active Subspace Initialization.** Active Subspace Initialization is most beneficial when activations exhibit pronounced low-rank structure, such as attention outputs, residual streams of some narrow datasets, or certain specialized situations. For activations without such low-rank behavior, the improvements are marginal (Appendix J.2).

## 7. Conclusion

We identified the low-rank structure of attention outputs as a fundamental property of Transformer models and a key cause of dead features in sparse dictionary learning. Our proposed *Active Subspace Initialization* method addresses this by aligning SAE features with the intrinsic geometry of activations, reducing dead features while improving reconstruction quality. The approach generalizes beyond SAEs to sparse replacement models.

## Impact Statement

This work aims to advance the understanding of transformer architectures and sparse dictionary learning in large language models. Our research is methodological in nature and does not raise any specific ethical concerns. While improvements in model efficiency and interpretability could indirectly influence the deployment of language models, we do not anticipate any direct negative societal consequences.

## Acknowledgements

This work was supported by the National Natural Science Foundation of China (No. 62236004).

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

| Layer | 3 | 7 | 11 | 15 | 19 | 23 | 27 | 31 |
|---|---|---|---|---|---|---|---|---|
| Sum of head ranks | **3601** | **3515** | **3456** | **3743** | **3712** | **3745** | **3723** | **3759** |
| Attention output rank | 2429 | 2351 | 2367 | 2506 | 2590 | 2656 | 2572 | 2012 |

*Table 1.* Comparison between **the sum of the effective ranks of individual attention heads** and *the effective rank of the full attention output* on representative layers. The remaining layers exhibit the same pattern, with the summed head ranks consistently exceeding the rank of the full attention output. It is the result of Llama-3.1-8B on SlimPajama (Soboleva et al., 2023) dataset.

## A. Activation Sources

The spectral characteristics of activations vary substantially across model architectures, datasets, and positional contexts. Below, we describe the experimental configurations used to support a broad and representative analysis.

**Models**   We study four large language models of different families–GPT-2[9], Llama-3.1-8B[10], Qwen3-8B[11], and Gemma-2-9B[12]–all based on the Transformer architecture. This allows us to assess the robustness of spectral properties under varying model training configurations.

**Datasets**   To investigate how dataset diversity affects activation spectra, we select two datasets with varying linguistic and domain characteristics: (1) SlimPajama[13], an English corpus comprising web text, Github, Arxiv and other sources. *Multi-Corpora* in Figure 2 denotes the setting where all SlimPajama components are jointly used with random mixing. *Github* and *Arxiv* correspond to the code and scientific paper subsets of SlimPajama, respectively. (2) CCI3-Data[14], a Chinese dataset with broad domain coverage, which is used in Appendix D as a supplement.

**Activation Positions**   We analyze three types of activations: (1) attention output, (2) MLP output, and (3) residual stream (post layer).

## B. Formal Definition of *Fraction of Loss Recovered*

Given the original language model cross-entropy loss is $loss_{\text{original}}$, the loss after ablating the activation at a specific position to zero is $loss_{\text{zero}}$, and the loss after replacing the original activation projected to the subspace spanned by first $n$ singular vectors is $loss_{\text{recovered}}$. Then, for these $n$ components, the fraction loss recovered is calculated as:

$$\frac{loss_{\text{zero}} - loss_{\text{recovered}}}{loss_{\text{zero}} - loss_{\text{original}}}$$

## C. Error analysis in Singular Value Decomposition

For the attention output of layer 15 of Llama-3.1-8B, we performed 5 times of singular value decompositions, using different 10 million tokens for each, and calculated the Coefficient of Variation (CV) for each singular value across these 5 runs. The maximum CV was only $4.9 \times 10^{-3}$, and the mean and standard deviation of the effective rank computed from these 5 SVD results were 2523.165 and 0.404, respectively, with a CV of $1.5 \times 10^{-4}$. These error experiments show that using 10 million tokens for singular value decomposition is sufficiently stable.

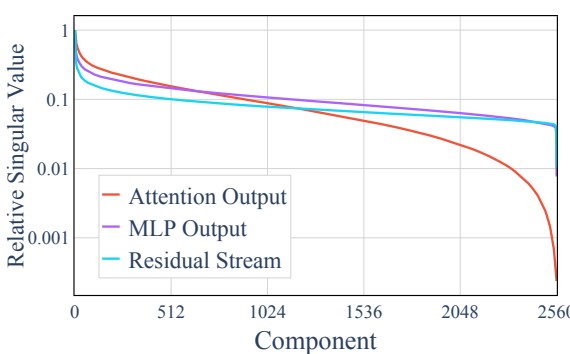

*(a)* Middle layer of pythia-2.8b; SlimPajama
Effective Dimensionality: Attention Output 1670;
MLP Output 2252; Residual Stream 2327

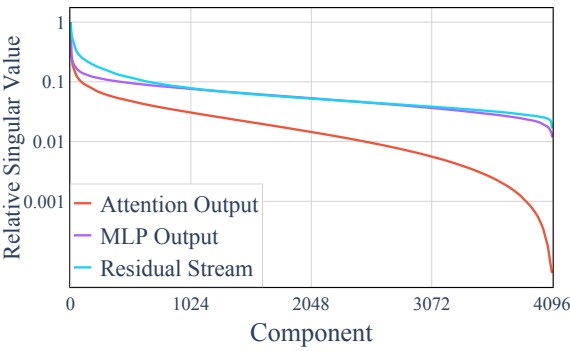

*(b)* Middle layer of Qwen3-8B; CCI3-Data
Effective Dimensionality: Attention Output 2356;
MLP Output 3558; Residual Stream 3140

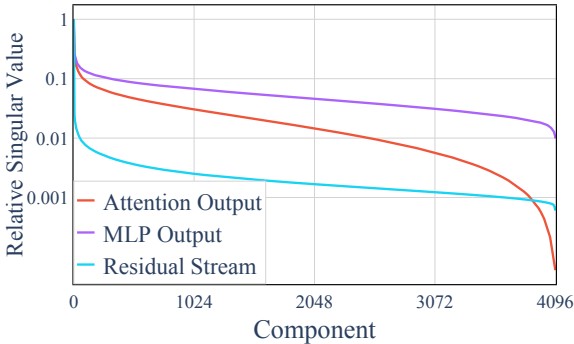

*(c)* Middle layer of Qwen3-8B; SlimPajamaGithub
Effective Dimensionality: Attention Output 2410;
MLP Output 3495; Residual Stream 2000

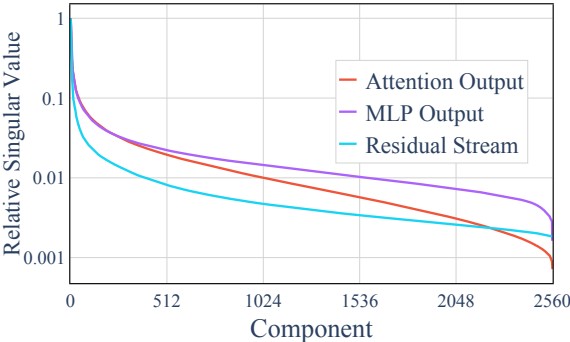

*(d)* Middle layer of Qwen3-4B; SlimPajama
Effective Dimensionality: Attention Output 1122;
MLP Output 1485; Residual Stream 990

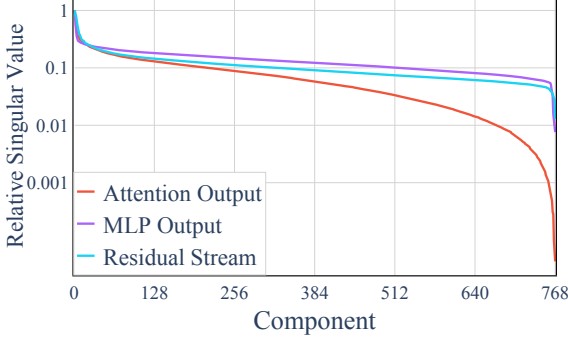

*(e)* Middle layer of gpt2; SlimPajama
Effective Dimensionality: Attention Output 515;
MLP Output 703; Residual Stream 656

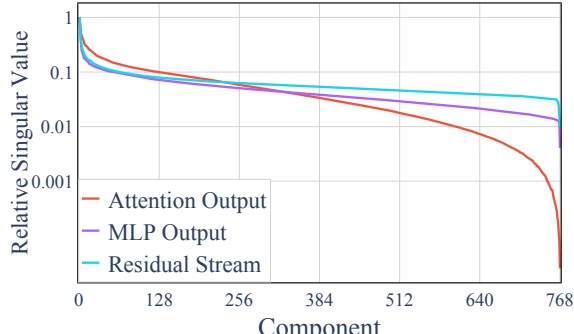

*(f)* Middle layer of pythia-160m; SlimPajama
Effective Dimensionality: Attention Output 434;
MLP Output 594; Residual Stream 662

*Figure 10*

## D. More Singular Spectrum and Effective Rank Results

### D.1. Across Models and Datasets

We present relative singular values for some other model-dataset pairs in Figure 10. Models include pythia-160m[15], pythia-2.8b[16]. Datasets include SlimPajamaGithub (subset of SlimPajama) and CCI3-Data.

### D.2. Across Layers and Activation Positions

We present the effective rank for activations that are commonly used to train SAEs in Figure 11, including **Concatenated Outputs of all Attention Heads** (Z), attention output, the hidden activations of MLP (post activation fuction), MLP output, residual stream. All effective ranks are computed on SlimPajama.

## E. SAE Training Details

We train SAEs as the following description.

### E.1. Hyperparameters

**Model, Dataset, Layer, Pos**   Llama-3.1-8B, SlimPajama, 15(index start at 0), attention output.

**Sparsity**   We empirically set $k = 50$ for a reasonable sparsity following He et al. (2024), except the experiments for sweeping $k$.

**Dictionary Size**   We empirically set $n_{features} = 32768$ which is $8 \times d_{model}$, except the experiments for sweeping dictionary size (scaling law).

**Batch Size**   We empirically set the batch size to 4096.

**Optimizer**   We use the Adam and SparseAdam optimizer, both with $\beta_1 = 0.9$, $\beta_2 = 0.999$, and $\epsilon = 10^{-8}$. Unless otherwise specified, Adam is used by default.

**Learning Rate**   The learning rate for **Adam** and **SparseAdam** is sweeped separately in [1e−5, 2e−5, 4e−5, 6e−5, 8e−5, 1e−4, 2e−4, 4e−4], and we ultimately use $4e-5$ for **Adam** and $6e-5$ for **SparseAdam**. We employ a three-phase learning rate schedule consisting of a linear warm-up, a stable phase, and a linear decay. The learning rate increases linearly from zero to its maximum value over the first 500 steps, remains constant during the intermediate phase, and then decays linearly to 1% of the maximum value over the final 20% of the total training steps.

**AuxK**   We follow Gao et al. (2024) to set auxiliary loss coefficient $\alpha$ as $\frac{1}{32}$. We sweep the $k_{aux}$ in [256, 512, 1024, 2048] and finally choose 512. We also sweep $\alpha$ and find the results are less sensitive to $\alpha$ in a reasonable interval.

**Dimension of Subspace for SAE Initialization** ($d_{init}$)   We use 768 for all experiments, except the experiments for sweeping $d_{init}$. We refer readers to Appendix J.3 for the reason.

**Total Tokens**   We use 800M tokens for each SAE training.

---

[9]https://huggingface.co/openai-community/gpt2
[10]https://huggingface.co/meta-llama/Llama-3.1-8B
[11]https://huggingface.co/Qwen/Qwen3-8B
[12]https://huggingface.co/google/gemma-2-9b
[13]https://huggingface.co/datasets/cerebras/SlimPajama-627B
[14]https://huggingface.co/datasets/BAAI/CCI3-Data
[15]https://huggingface.co/EleutherAI/pythia-160m
[16]https://huggingface.co/EleutherAI/pythia-2.8b

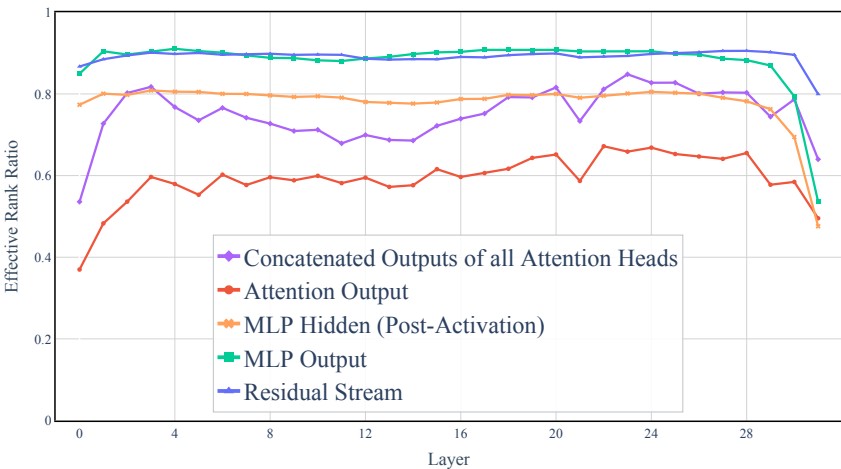

*(a)* The number of the effective rank of each activation in Llama-3.1-8B

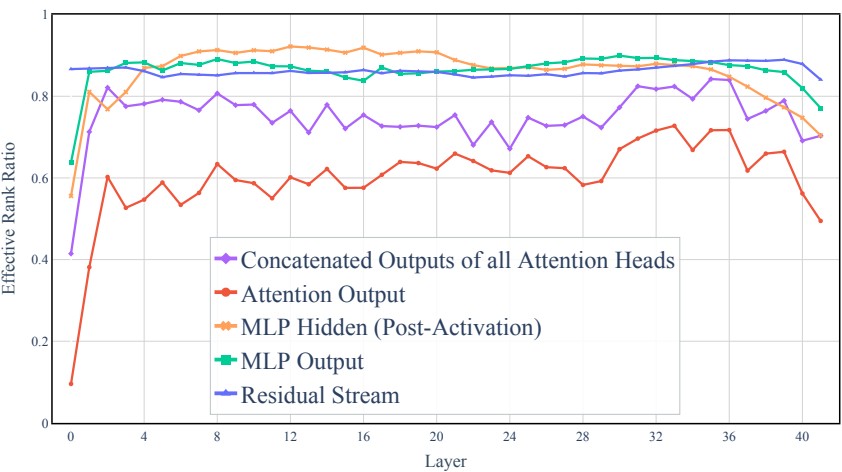

*(b)* The number of the effective rank of each activation in Gemma-2-9B

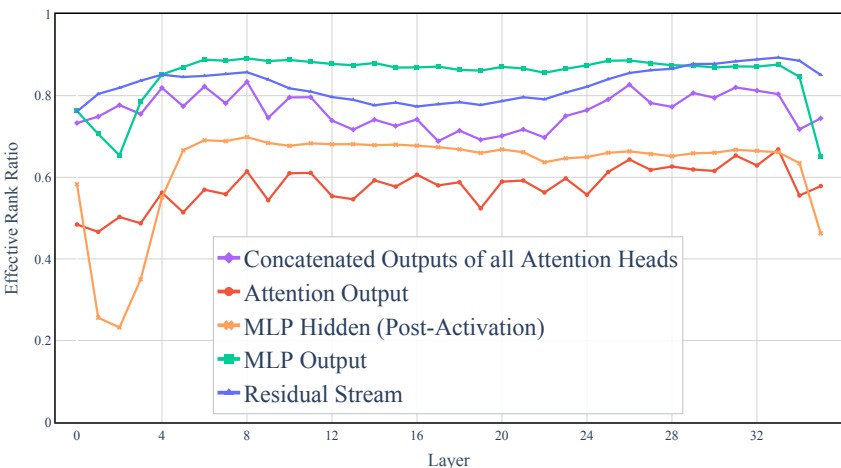

*(c)* The number of the effective rank of each activation in Qwen3-8B

*Figure 11*

### E.2. Collecting Activations

We truncate each document to 1024 tokens and prepend a <bos> token to the beginning of each document. During training, we exclude the activations corresponding to the <bos> , <eos>and <pad> tokens.

It has been observed that activations from different sequence positions within the same document are often highly correlated and may lack diversity. To mitigate this issue, it is common to introduce randomness into the training data. Our shuffling strategy maintains a buffer that is reshuffled whenever the buffer is refilled.

### E.3. Initialization

The decoder columns $W^{dec}_{:,i}$ are initialized uniformly, and the optimal norm for them is found through a grid search to minimize the initial reconstruction loss. We find that the specific initialization norm has little impact, as long as in a reasonable scope. For example, initializing $W^{dec}_{:,i}$ uniformly with a fixed bound, as in Conerly et al. (2025), yields similar results. The encoder weights $W^{enc}$ are initialized as the transpose of $W^{dec}$, while both the encoder bias $b^{enc}$ and decoder bias $b^{dec}$ are set to zero.

### E.4. Jumprelu SAEs

We trained JumpReLU SAEs (Rajamanoharan et al., 2024b) under two distinct hyperparameter configurations: one maintaining a consistently low $\ell_0$ value throughout training, and another where $\ell_0$ is gradually decreased from a higher initial value. Unless otherwise specified, all JumpReLU SAEs were trained using the same settings as Conerly et al. (2025), which corresponds to the latter configuration. The key modifications for the former setting are as follows: (1) we initialized the encoder bias to zero instead of applying the heuristic that equalizes feature activation counts at initialization, and (2) we kept the sparsity coefficient fixed rather than employing a global warm-up schedule. As a result, the $\ell_0$ sparsity level started at a relatively low value early in training. This design is critical to our approach: we observed that if the model remains in a high-$\ell_0$ regime (e.g., on the order of $d_{\text{model}}/2$) for an extended period before sparsity increases, the feature directions tend to drift away from the active subspace during this phase, thereby diminishing the effectiveness of our method (Appendix L.1).

## F. Additional Analysis on Effective Dimensionality and Dead Features

This section provides additional experimental evidence supporting the claim that *low effective dimensionality strongly correlates with a higher proportion of dead features* in sparse autoencoders (SAEs). In Figure 5, we report the effective rank of the residual stream, attention output, and MLP output at each layer of `Llama-3.1-8B`, along with the proportion of dead features in the SAEs trained on these activations. All SAEs were obtained from `Llamascope` (He et al., 2024), which uses the same dictionary size (32768 features) and sparsity level ($L_0 = 50$).

To provide systematic analysis, we additionally train SAEs across multiple dictionary sizes (16384, 32768, 65536) and sparsity levels ($L_0 \in \{32, 64, 128\}$) following Appendix E The SAEs are trained on the activations of `Llama-3.1-8B`, and the corresponding effective ranks of these activations can be found in Figure 4 of the main paper. Tables 2–4 summarize the proportion of dead features across these configurations.

Across all settings, attention outputs consistently show substantially higher dead-feature ratios than residual streams. This trend holds even when the dictionary size varies by a factor of four and the sparsity level varies by a factor of four.

*Table 2.* Proportion of dead features for $L_0 = 32$ across different dictionary sizes.

| Activation (Effective Rank) | 16384 | 32768 | 65536 |
|---|---|---|---|
| Layer 7 attention (2351) | **84.80%** | **90.31%** | **94.02%** |
| Layer 7 residual (3664) | 1.61% | 6.69% | 17.10% |
| Layer 15 attention (2506) | **68.70%** | **79.86%** | **87.22%** |
| Layer 15 residual (3611) | 27.01% | 45.70% | 61.33% |
| Layer 23 attention (2654) | **58.45%** | **70.48%** | **78.81%** |
| Layer 23 residual (3634) | 0.13% | 0.26% | 1.35% |

*Table 3.* Proportion of dead features for $L_0 = 64$ across different dictionary sizes.

| Activation (Effective Rank) | 16384 | 32768 | 65536 |
|---|---|---|---|
| Layer 7 attention (2351) | **66.97%** | **75.65%** | **82.96%** |
| Layer 7 residual (3664) | 0.02% | 0.06% | 0.15% |
| Layer 15 attention (2506) | **41.65%** | **54.68%** | **67.43%** |
| Layer 15 residual (3611) | 1.73% | 7.96% | 18.83% |
| Layer 23 attention (2654) | **41.58%** | **56.70%** | **67.05%** |
| Layer 23 residual (3634) | 0.15% | 0.09% | 0.08% |

*Table 4.* Proportion of dead features for $L_0 = 128$ across different dictionary sizes.

| Activation (Effective Rank) | 16384 | 32768 | 65536 |
|---|---|---|---|
| Layer 7 attention (2351) | **49.85%** | **56.96%** | **64.83%** |
| Layer 7 residual (3664) | 0.00% | 0.00% | 0.02% |
| Layer 15 attention (2506) | **15.09%** | **25.61%** | **37.11%** |
| Layer 15 residual (3611) | 0.07% | 0.12% | 0.44% |
| Layer 23 attention (2654) | **21.90%** | **39.02%** | **52.68%** |
| Layer 23 residual (3634) | 0.14% | 0.09% | 0.07% |

## G. Complete Results of SAE metrics

We use 3 different random seeds for all experiments in Figure 7 and compute the mean values and the standard deviations of each metrics, as shown in Table 5.

## H. Statistical Significance Test

To assess whether the performance improvements introduced by ASI are statistically significant, we conducted a comprehensive significance analysis across multiple evaluation metrics.

### H.1. Experimental Setup

We evaluate the statistical significance of performance differences between ASI and two baseline methods (TopK and AuxK) under the following controlled setting:

- **Model / Layer / $L_0$ / Dictionary Size:** Llama-3.1-8B, Layer 15, $L_0 = 50$, dictionary size = 32,768.

- **Number of runs:** 15 independent trials for each method, each with a different random seed.

- **Evaluation metrics:** Dead Feature Count, Normalized MSE, and $\Delta$ LM Loss. All metrics follow a "lower is better" criterion.

- **Comparisons performed:** ASI vs. TopK and ASI vs. AuxK for all three metrics.

### H.2. Hypothesis Testing Framework

For each metric and each baseline method, we perform Welch's t-test (also known as Welch's unequal-variance t-test), which does not assume equal variances between groups. For ASI and a given baseline method, we test the following hypotheses:

$$H_0 : \mu_{\text{ASI}} \geq \mu_{\text{baseline}} \quad \text{(ASI is worse than or equal to the baseline)},$$
$$H_1 : \mu_{\text{ASI}} < \mu_{\text{baseline}} \quad \text{(ASI outperforms the baseline)}.$$

This is a one-tailed test, as we explicitly test whether ASI achieves significantly lower metric values.

*Table 5.* Comparison of Base, AuxK, and ASI across different $L_0$ settings. Numbers show mean $\pm$ std over 3 seeds.

| $L_0$ | Metric | Base | AuxK | ASI |
|---|---|---|---|---|
| 40 | Dead Feature | $20395.00 \pm 72.77$ | $37.00 \pm 5.20$ | $\mathbf{10.67 \pm 1.15}$ |
| | Normalized MSE | $0.36323 \pm 0.00018$ | $0.34328 \pm 0.00011$ | $\mathbf{0.33724 \pm 0.00022}$ |
| | Delta LM Loss ($\times 10^{-3}$) | $9.650 \pm 0.040$ | $8.801 \pm 0.053$ | $\mathbf{8.697 \pm 0.033}$ |
| 50 | Dead Feature | $16144.33 \pm 129.52$ | $54.67 \pm 5.51$ | $\mathbf{4.00 \pm 1.73}$ |
| | Normalized MSE | $0.33367 \pm 0.00014$ | $0.32241 \pm 0.00020$ | $\mathbf{0.31680 \pm 0.00006}$ |
| | Delta LM Loss ($\times 10^{-3}$) | $8.375 \pm 0.029$ | $7.958 \pm 0.032$ | $\mathbf{7.847 \pm 0.050}$ |
| 60 | Dead Feature | $12239.33 \pm 165.51$ | $76.00 \pm 9.17$ | $\mathbf{2.67 \pm 1.53}$ |
| | Normalized MSE | $0.31106 \pm 0.00007$ | $0.30555 \pm 0.00007$ | $\mathbf{0.30000 \pm 0.00005}$ |
| | Delta LM Loss ($\times 10^{-3}$) | $7.503 \pm 0.059$ | $7.325 \pm 0.017$ | $\mathbf{7.214 \pm 0.026}$ |
| 70 | Dead Feature | $8854.67 \pm 40.51$ | $115.33 \pm 15.37$ | $\mathbf{1.67 \pm 0.58}$ |
| | Normalized MSE | $0.29295 \pm 0.00008$ | $0.29064 \pm 0.00008$ | $\mathbf{0.28575 \pm 0.00013}$ |
| | Delta LM Loss ($\times 10^{-3}$) | $6.805 \pm 0.074$ | $6.694 \pm 0.021$ | $\mathbf{6.664 \pm 0.066}$ |
| 80 | Dead Feature | $6311.33 \pm 55.77$ | $109.67 \pm 11.50$ | $\mathbf{1.00 \pm 1.73}$ |
| | Normalized MSE | $0.27787 \pm 0.00003$ | $0.27715 \pm 0.00003$ | $\mathbf{0.27341 \pm 0.00003}$ |
| | Delta LM Loss ($\times 10^{-3}$) | $6.259 \pm 0.005$ | $6.217 \pm 0.028$ | $\mathbf{6.168 \pm 0.033}$ |

We use the following SciPy function for all tests:

```
scipy.stats.ttest_ind(ASI, baseline, equal_var=False, alternative='less').
```

### H.3. Results

Table 6 reports the resulting p-values for all comparisons. A smaller p-value indicates stronger evidence that ASI outperforms the baseline.

*Table 6.* Welch's t-test results for ASI compared with TopK and AuxK across 15 random seeds.

| Comparison | Metric | p-value |
|---|---|---|
| ASI vs. TopK | Dead Feature Count | $3.26 \times 10^{-35}$ |
| | Normalized MSE | $2.99 \times 10^{-40}$ |
| | $\triangle$ LM Loss | $1.14 \times 10^{-23}$ |
| ASI vs. AuxK | Dead Feature Count | $4.33 \times 10^{-15}$ |
| | Normalized MSE | $6.33 \times 10^{-40}$ |
| | $\triangle$ LM Loss | $1.27 \times 10^{-6}$ |

Across all evaluation metrics and both baseline methods, the p-values are far below standard significance thresholds (e.g., $\alpha = 0.05$). Therefore, we reject the null hypothesis for all comparisons. These results demonstrate that the improvements achieved by ASI are statistically significant and robust across random seeds.

## I. Additional Evaluation Across Layers, Models, and Datasets

To assess the robustness and generality of ASI, we extend our experiments beyond the primary configuration used in the main paper (Llama-3.1-8B, Layer 15, SlimPajama). In particular, we investigate whether the advantages of ASI over baseline approaches (TopK and AuxK) persist across different layers, models, and datasets. We consider this evaluation essential, as mechanisms in sparse autoencoding can vary substantially across architectural depth, data distribution, and model family.

### I.1. Evaluation on Llama-3.1-8B Across Multiple Layers

We first evaluate ASI on two additional layers of Llama-3.1-8B (Layers 7 and 23), using the SlimPajama dataset. Activations are taken from the attention output. Results are summarized in Table 7.

We observe that ASI consistently achieves the lowest reconstruction error (Normalized MSE) and the smallest degradation in language modeling performance ($\triangle$ LM loss). For Layer 7, ASI still retains a number of dead features, which may be

*Table 7.* Performance of ASI and baseline methods on Llama-3.1-8B Layers 7 and 23. Lower values indicate better performance.

| Layer | Dead Feature Count | | | Normalized MSE | | | $\Delta$ LM Loss ($\times 10^{-3}$) | | |
|---|---|---|---|---|---|---|---|---|---|
| | Base | AuxK | ASI | Base | AuxK | ASI | Base | AuxK | ASI |
| 7 | 25836 | 98 | **27** | 0.32870 | 0.29800 | **0.28882** | 5.414 | 4.589 | **4.430** |
| 23 | 15542 | **332** | 5308 | 0.21302 | 0.20235 | **0.20141** | 1.942 | 1.865 | **1.845** |

attributed to its smaller effective rank (2351) compared to Layer 23 (2654). Since we use a fixed $d_{\text{init}}$ across layers, this mismatch can lead to remaining dead features. Despite this, ASI still achieves the lowest MSE on both layers.

### I.2. Evaluation on Qwen3-8B Across Layers and a New Dataset

To further test cross-model and cross-dataset generality, we conduct experiments on the Qwen3-8B model using the fineweb-edu dataset, again using attention-output activations. Results for Layers 8 and 26 are shown in Table 8.

*Table 8.* Performance comparison on Qwen3-8B Layers 8 and 26 using the fineweb-edu dataset. Lower values indicate better performance.

| Layer | Dead Feature Count | | | Normalized MSE | | | $\Delta$ LM Loss ($\times 10^{-3}$) | | |
|---|---|---|---|---|---|---|---|---|---|
| | Base | AuxK | ASI | Base | AuxK | ASI | Base | AuxK | ASI |
| 8 | 19048 | **4** | 7 | 0.31228 | 0.28849 | **0.28533** | 2.9561 | 2.5911 | **2.5667** |
| 26 | 16286 | **66** | 566 | 0.30090 | 0.28127 | **0.28087** | 1.3406 | 1.2980 | **1.2922** |

The results again confirm that ASI achieves the lowest reconstruction error and the smallest increase in LM loss across both layers. The near-dead-feature-free representation produced by AuxK is also observed, but ASI consistently outperforms AuxK in reconstruction quality and LM preservation.

### I.3. Summary

Across all tested configurations—spanning multiple layers, two large language model families, and two datasets—ASI exhibits consistent advantages over baseline methods. These evaluations provide strong empirical evidence that the benefits of ASI are not confined to a specific layer, model, or dataset, but instead generalize across diverse settings.

## J. Ablation Study

### J.1. Active Subspace Init vs Random Subspace Init

We employ random subspace initialization as a baseline and observe that it consistently degrades SAE training across all metrics, as shown in Figure 12.

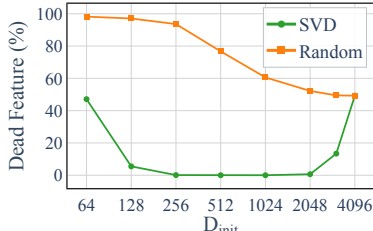 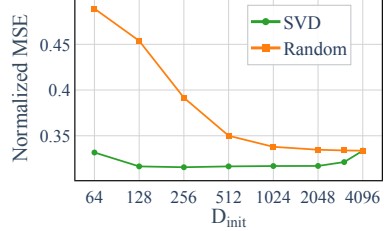 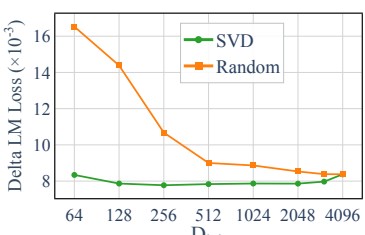

*Figure 12.* For activations with a full space dimension of 4096, proportion of dead features (left), normalized MSE (mid) and Delta LM loss (right) across different subspace dimensions. Random subspaces are used as the baseline, whereas only initialization with the active subspace yields improvement.

## J.2. Apply on Near-Full-Rank Activation

We also apply Active Subspace Initialization (ASI) to near-full-rank activations, such as those in the residual stream, to evaluate its generality. When training an SAE on the post-layer-15 residual stream of Llama-3.1-8B, we find ASI yields minimal gains (Figure 13). This is consistent with our expectation, as these activations inherently exhibit a lower rate of dead features even with standard initialization.

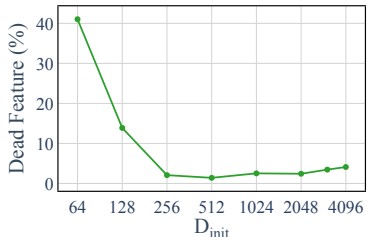 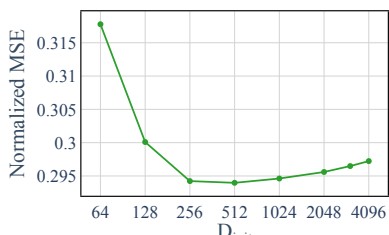 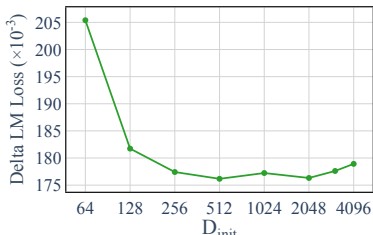

*Figure 13.* For activations with a full space dimension of 4096, proportion of dead features (left), normalized MSE (mid) and Delta LM loss (right) across different subspace dimensions. Random subspaces are used as the baseline, whereas only initialization with the active subspace yields improvement.

## J.3. Choice of the Initial Dictionary Size $d_{init}$

As shown in Figure 6, $d_{init}$ is a hyperparameter with a wide range of acceptable values (from 256 to 2048). We hypothesize that the performance degradation when $d_{init}$ is very low is due to a combination of the dictionary failing to cover the subspace containing the key information needed for effective reconstruction and the features being too crowded.

# K. Pseudo-code for implementing Active Subspace Init

Below is a PyTorch-style pseudo-code for Active Subspace Initialization.

**Use on SAE**

```
# X: activation batch [batch_size, d_model]
# W_E: decoder weight [d_model, d_sae]
# W_D: decoder weight [d_sae, d_model], initialized uniformly
# d_active_subspace: target subspace dimension

# 1. Demean the activations
demeaned_X = X - X.mean(dim=0) # [batch_size, d_model]

# 2. Compute SVD
U, S, V = torch.svd(demeaned_label) # V: [d_model, d_model]

# 3. Take top-d_init singular vectors
proj_weight = V[:, :d_init]  # [d_model, d_init]

# 4. Fold projection into decoder weights
W_D.copy_(W_D @ proj_weight @ proj_weight.T)

# 5. Init W_E with W_D.T
W_E.copy_(W_D.T)
```

**Use on Lorsa**

```
# Input:
# X: input activations [b, s, d]   (b=batch_size, s=seq_len, d=d_model)
```

```
# mhsa: pretrained MHSA module
#   mhsa.W_V: [n, d, h]   (n=n_heads, h=d_head, note: n*h=d)
#   mhsa.W_O: [n, h, d]
# Lorsa parameters to initialize:
#   W_V: [n_lorsa, d]   (n_lorsa = number of Lorsa heads)
#   W_O: [n_lorsa, d]
# d_qk: Lorsa head dimension for initialization

# 1. Compute per-head V projections
X_flat = X.reshape(b*s, d)                          # [b*s, d]
W_V_cat = mhsa.W_V.permute(1,0,2).reshape(d, d)     # [d, d]
V_per_head = (X_flat @ W_V_cat).reshape(b*s, n, h)  # [b*s, n, h]

# 2. Project V back to d_model space for each head
# captured_v[:, i, :] = V_per_head[:, i, :] @ mhsa.W_V[i].T
captured_v = einsum('bnh, nhd -> bnd',
                    V_per_head, mhsa.W_V.permute(0,2,1))
# captured_v: [b*s, n, d]

# 3. Initialize Lorsa heads from each original head's active subspace
rate = n_lorsa // n
for i in range(n):
    slice_i = [rate*i : rate*(i+1)]

    # 3.1 Extract this head's captured V
    v = captured_v[:, i, :]              # [b*s, d]

    # 3.2 Demean
    demeaned_v = v - v.mean(dim=0)       # [b*s, d]

    # 3.3 SVD on transposed data to get principal directions
    U, S, _ = svd(demeaned_v.T)          # demeaned_v.T: [d, b*s]
                                         # U: [d, d]

    # 3.4 Take top-d_qk principal directions as projection
    proj = U[:, :d_qk]                   # [d, d_qk]

    # 3.5 Update W_V: project from initial d_qk space to principal subspace
    W_V[slice_i] = W_V[slice_i, :d_qk] @ proj.T    # [rate, d]

    # 3.6 Update W_O: chain updated W_V through original head's OV circuit
    # OV_i = mhsa.W_V[i] @ mhsa.W_O[i]: [d, h] @ [h, d] = [d, d]
    W_O[slice_i] = W_V[slice_i] @ mhsa.W_V[i] @ mhsa.W_O[i]
    # [rate, d] @ [d, h] @ [h, d] = [rate, d]

# 4. Normalize all Lorsa weights (row-wise)
W_V = W_V / W_V.norm(dim=1, keepdim=True)   # [n_lorsa, d]
W_O = W_O / W_O.norm(dim=1, keepdim=True)   # [n_lorsa, d]
```

The strategy of initialize $W_O$ in Lorsa is a method like the **tied initialization** used in SAEs to ensure alignment between feature encoding and decoding[17]. This approach has been shown to be crucial for reducing dead features in SAEs (Gao et al., 2024). We think the same thought could also be used to improve the replacement model for MLP (trancoder and cross layer transcoder), which we leave a deeper investigation to future work.

---

[17]"Match" means encoder can be initialized to predict relatively accurate feature activation values for decoder.

# L. Use ASI on other Activation Functions

## L.1. Jumprelu

Another wildly used activation fuction is Jumprelu (Rajamanoharan et al., 2024b). We trained the Jumprelu SAEs under two different hyperparameter settings: one with a consistently low $\ell_0$ value and another where $\ell_0$ gradually decreases from a higher initial value, as described in Appendix E.4. We observed that our method is effective in the former case (Figure 14) but shows little improvement in the latter (Figure 15). We train these SAEs following Appendix E.

For cases where one follows a schedule that gradually reduces $\ell_0$ from a high initial value, we recommend first applying PCA to reduce the dimensionality of the data. The SAE can then be trained on the reduced representation until the $\ell_0$ level reaches the target range. Afterwards, the PCA projection matrix can be folded into the model parameters, and training can continue in the original space. This achieves a similar effect without the drawbacks of prolonged training in the high-$\ell_0$ regime.

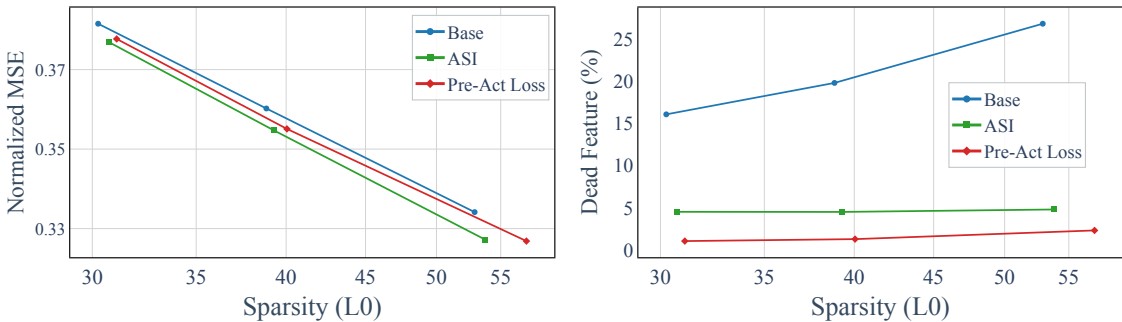

Figure 14. For the attention output of the middler layer of Llama-3.1-8B, using ASI on JumpRelu SAEs which has a low initial $\ell_0$ is effective. Details in Appendix L.1

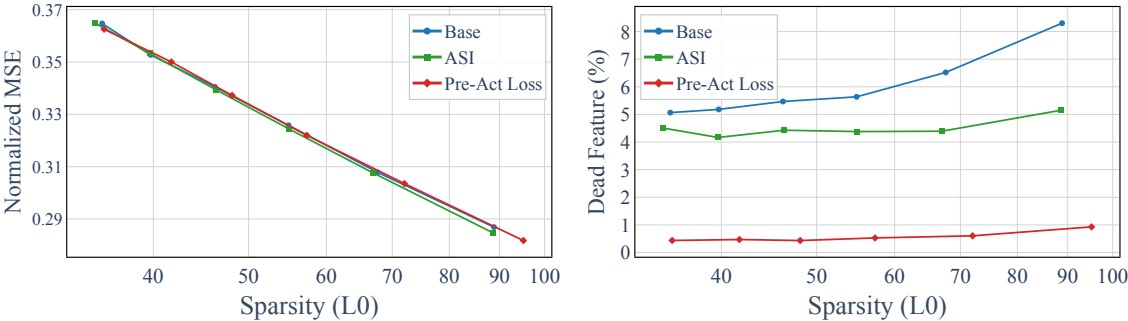

Figure 15. For the attention output of the middler layer of Llama-3.1-8B, using ASI on JumpRelu SAEs which has a high initial $\ell_0$ than gradually decreasing shows little improvement. Details in Appendix L.1

## L.2. TopK with K Anneal

To enhance the finding in Section L.1, we conduct experiments on a variant of TopK, which sets K to a high value and then lets it decrease during training (He et al., 2024). We find ASI also fails in this case (Figure 16).

# M. Stale Momentum as Another Root Cause of Dead Features

Recent work by Bricken et al. (2023a) identifies *stale momentum* as a key cause to dead feature formation. Specifically, when a feature remains inactive over training steps, its associated optimizer momentum continues to accumulate. If the feature activates, the stale momentum results in disproportionately large updates, destabilizing training and potentially suppressing that feature permanently.

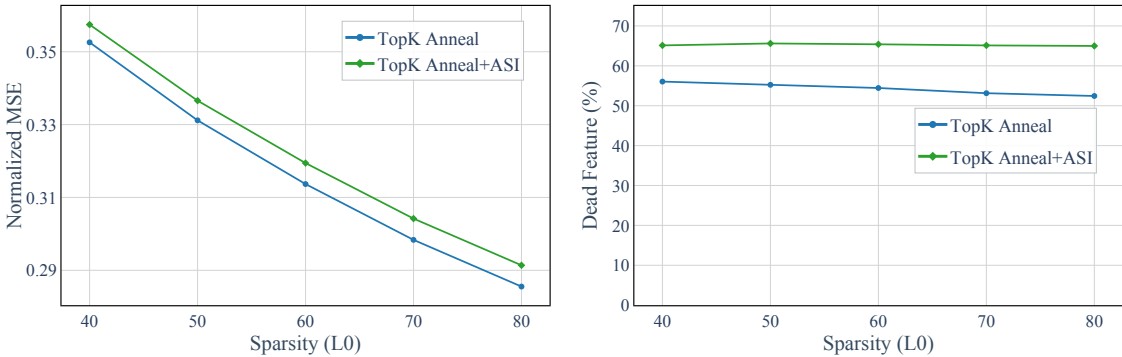

*Figure 16.* For the attention output of the middler layer of Llama-3.1-8B, using ASI on TopK SAEs which sets K to a high value and then lets it decrease during training fails. Details in Appendix L.2

To directly address this, we adopt *SparseAdam*, an optimizer tailored for sparse activation settings, designed for more efficient use of compute and memory. SparseAdam updates both parameters and moments only when the corresponding feature is active. This could effectively prevent the harmful accumulation of stale momentum. Empirically, we observe that this change substantially reduces the rate of dead feature formation in large-scale SAE training. We believe that this is a core technique for scaling sparse dictionary methods, as stale momentum is a common problem for them.

## N. Compare Features of SAEs with and without ASI

### N.1. Monosemanticity

We conducted an additional analysis to assess the degree of monosemanticity exhibited by features learned by the base TopK SAE and the ASI-enhanced SAE. Following the rubric of Cunningham & Conerly (2024), we performed a blinded evaluation of 100 randomly sampled features and recorded the semantic consistency scores assigned to each feature.

For clarity, we reproduce below the scoring rubric used for evaluating activation consistency:

- **5**: Clear pattern with no deviating examples

- **4**: Clear pattern with one or two deviating examples

- **3**: Clear overall pattern but quite a few examples not fitting that pattern

- **2**: Broad consistent theme but lacking structure

- **1**: No discernible pattern

To provide transparency, Tables 9 summarize the distribution of scores for TopK and ASI.

Across both models, we found no statistically significant differences in score distributions at this scale of analysis. Variations between the two SAE variants were marginal and did not indicate systematic differences in feature quality. This result is consistent with our expectations, as our method does not modify the SAE architecture and is not designed to intervene in how features are formed. Consequently, the potential risk of degrading feature quality remains low.

### N.2. Analysis of SAE Features in the Dead Subspace

To assess whether ASI alters the SAE's behavior in directions corresponding to the dead subspace, we perform a comparative analysis between Attention Output SAEs trained with and without ASI under identical configurations (same number of features, $K$, and all other hyperparameters).

**Feature alignment with the dead subspace.** We compute the cosine similarity between each SAE feature and the dead subspace, restricting the analysis to alive features (which accounts for the difference in total counts). The distribution of cosine values is summarized in Table 10. Across all intervals, the ASI-initialized SAE exhibits a larger number of alive

*Table 9.* Monosemanticity score distributions for ASI-enhanced SAE and base TopK SAE features.

*(a)* ASI-enhanced SAE

| Score | Count |
|-------|-------|
| 5 | 17 |
| 4 | 14 |
| 3 | 6 |
| 2 | 5 |
| 1 | 8 |

*(b)* Base TopK SAE

| Score | Count |
|-------|-------|
| 5 | 18 |
| 4 | 12 |
| 3 | 6 |
| 2 | 6 |
| 1 | 8 |

features, while both methods exhibit very few features that align closely with the dead subspace. This suggests two possible explanations: (i) features in the dead subspace have extremely small magnitude and provide insufficient signal for the SAE to learn, or (ii) the dead subspace does not contain meaningful standalone features, and only small components of features reside in this region.

*Table 10.* Distribution of cosine similarity between SAE features and the dead subspace (alive features only).

| Method | [0.0, 0.05) | [0.05, 0.1) | [0.1, 0.15) | [0.15, 1] |
|--------|-------------|-------------|-------------|-----------|
| TopK | 10176 | 189 | 6 | 0 |
| ASI | 24576 | 936 | 8 | 0 |

**Reconstruction error in the dead subspace.**    We further project the reconstruction error onto the dead subspace to quantify the SAE's reconstruction performance on components lying in this region. As shown in Table 11, the reconstruction errors are nearly identical between the two methods, with the ASI showing only a marginal improvement.

*Table 11.* Reconstruction error projected onto the dead subspace.

| Method | MSE in dead subspace |
|--------|---------------------|
| TopK | 0.00350 |
| ASI | 0.00334 |

Overall, these analyses indicate that ASI has minimal impact on the SAE's behavior within the dead subspace, while substantially reducing the number of dead features.

## O. Lorsa Implementation Details

### O.1. Hyperparameters

**Model, Dataset, Layer**    Llama-3.1-8B, SlimPajama, 15(index start at 0).

**Dictionary Size**    We empirically set the num of Lorsa heads $n_{heads}^{Lorsa} = 32768$ which is $8 \times d_{model}$. We set the num of QK group of Lorsa $n_{qk}^{Lorsa} = 256$ which is $8 \times n_{heads}^{MHSA}$. We set the dimension of QK of Lorsa $d_{qk}^{Lorsa} = 128$ which is $d_{qk}^{MHSA}$.

**Batch Size**    We empirically set the batch size to 32768.

**Optimizer**    We use the Adam optimizer, with $\beta_1 = 0.9$, $\beta_2 = 0.999$, and $\epsilon = 10^{-8}$.

**Learning Rate**    The learning rate is sweeped in [1e−5, 2e−5, 4e−5, 6e−5, 8e−5, 1e−4, 2e−4, 4e−4], and we ultimately use 2e−4. We employ a three-phase learning rate schedule consisting of a linear warm-up, a stable phase, and a linear decay. The learning rate increases linearly from zero to its maximum value over the first 500 steps, remains constant during the intermediate phase, and then decays linearly to 1% of the maximum value over the final 20% of the total training steps.

**AuxK**   We follow Gao et al. (2024) to set auxiliary loss coefficient $\alpha$ as $\frac{1}{32}$. We sweep the $k_{aux}$ in [256, 512, 1024, 2048] and finally choose 512.

**Dimension of Subspace for SAE Initialization** ($d_{init}$)   Because the active subspace of the input and output of each MHSA heads is very close to the dimension of MHSA head ($d_{head}$), we set it directly to $d_{head}$. We found that increasing or decreasing this value did not improve performance.

**Total Tokens**   We use 800M tokens for each Lorsa training.

**Sequence Length**   We truncate each document to 2048 tokens. During training, we exclude the activations corresponding to the <bos> , <eos>and <pad> tokens.

## O.2. Initialization

We initialize the query and key matrices $W_Q$ and $W_K$ using Xavier uniform initialization (Glorot & Bengio, 2010). The value matrix $W_V$ is initialized from a normal distribution $\mathcal{N}(0, 1/\sqrt{d_{sae}})$, while the output matrix $W_O$ is initialized from $\mathcal{N}(0, 1/\sqrt{d_{model}})$. All bias terms $b_Q, b_K, b_V$, and $b_D$ (if used) are initialized to zero.

# P. Preliminary Attempts to Mitigate Attention Low-Rankness

Our main results suggest that the low-rank structure of attention outputs is a robust property of Transformer-based language models. A natural question is whether this low-rankness can be directly mitigated to improve model capacity or expressivity. We conducted several preliminary explorations along this direction. Overall, these attempts provide initial evidence that mitigating attention low-rankness is nontrivial: some architectural choices appear to have a modest effect, while directly optimizing for higher rank does not yield clear downstream benefits.

**Adjusting attention hyperparameters.**   We first examined whether attention hyperparameters affect the effective rank of attention outputs. Gemma-2-9B uses a different attention configuration from Llama-3.1-8B and Qwen-3-8B, and exhibits a slightly higher effective rank of attention outputs, approximately 62% of the full space compared to roughly 60% in the latter models. This suggests that hyperparameter scaling may influence attention-output rank, but the effect appears limited in magnitude.

**Architectural modifications.**   We next considered architectural variants that may reduce attention low-rankness. Gated attention introduces additional nonlinearity into the attention block and may therefore help mitigate rank collapse. To test this hypothesis, we evaluated public gated-attention models under matched settings. Compared with standard attention, gated attention yields a modest increase in effective rank, from approximately 59% to 63%. We also observe that models without *grouped-query attention* (GQA), such as GPT-2 and Pythia, tend to exhibit higher effective rank than GQA-based models such as Llama-3.1 and Qwen-3, with effective ranks of roughly 68% versus 60%, respectively. These results suggest that architectural choices can affect the severity of attention low-rankness, but any such modification must be weighed against efficiency and scalability trade-offs.

**Rank-targeting auxiliary loss.**   We further experimented with adding an auxiliary pretraining loss that explicitly encourages higher-rank attention outputs. Although this objective increases the measured effective rank, it does not improve cross-entropy loss and substantially increases training time. This indicates that naively optimizing rank as an auxiliary objective may not translate into better language-modeling performance.

**Gradient analysis.**   To better understand why rank-targeting objectives are ineffective, we analyzed gradients along the singular directions of attention outputs. We find that directions associated with near-zero singular values often have the largest gradient norms. This suggests that low-rankness is not simply caused by a lack of optimization signal in low-variance directions. Instead, the model appears to receive gradient pressure along these directions but does not convert it into persistently high-variance activations, which may explain why directly increasing rank through auxiliary losses yields limited benefits.

Taken together, these preliminary results suggest that attention low-rankness is difficult to mitigate through simple hyperparameter changes or direct rank regularization. Architectural modifications such as gated attention appear more promising,

but their gains remain modest and must be considered together with computational efficiency. We leave a systematic study of rank-preserving attention architectures to future work.

