# OpenReview forum: "Dimensional Collapse in Transformer Attention Outputs: A Challenge for Sparse Dictionary Learning"
_ICML.cc/2026/Conference — ICML 2026 regular_

### Official Review · Reviewer_iM9h · 2026-03-07

**Soundness:** 4
**Presentation:** 3
**Significance:** 3
**Originality:** 4
**Overall Recommendation:** 5
**Confidence:** 4

**Summary:**

The paper introduces a new method called Active Subspace Initialization (ASI) for initializing sparse autoencoder weights aligned with the low-dimensional structure of network outputs. To motivate this method, they first study the effective ranks of pretrained LLMs and show that they are markedly lower for the output of the attention block relative to any other place in the network, and give mechanistic explanations for this according to the spectra of weight/activation matrices.  Then they introduce the ASI method which aligns sparse autoencoder dictionaries with this low-rank subspace of attention outputs. Experiments show that this method sharply reduces the number of "dead features" and improves performance across several metrics.

**Compliance With Llm Reviewing Policy:**

Affirmed.

**Final Justification:**

My initial review was positive and the rebuttal answered remaining questions satisfactorily, so I continue to recommend acceptance.

**Key Questions For Authors:**

- How do you computationally detect dead features?
- Does the initial batch size of the activations you use for the low dimensional subspaces matter wrt the quality of the learned model? Can you measure how batch size scaling affects the quality of the trained models?
- What is the initialization cost relative to a single forward pass or the regular initialization (or similar, just to get understanding of how expensive the initialization is)?

**Limitations:**

yes

**Strengths And Weaknesses:**

**Strengths**
- The motivation and "story" is clear throughout the paper, and the initial study of ranks and spectra illustrate the intuition well. From this perspective, the ASI method is absolutely natural and it makes sense why it would perform well.
- The empirical performance of the ASI method is often effective at reducing the number of dead features: for a one-time cost at initialization of the SAE, it outperforms popular baselines (manual Top-K sparsity and sparsity enforced by an auxiliary loss), greatly reducing the proportion of dead features.
- The extension to more general sparse/low-rank distillation methods further highlights the applicability of the idea.

**Weaknesses**
- Despite drastically reducing the number of dead features in some cases, ASI provides only marginal (albeit consistent) improvement to other metrics such as sparsity in these same settings.
- The improvement on # of dead features may be a bit volatile; for example in Table 7 the number of dead features at layer 26 SAE is around 8x as high as the auxiliary loss baseline, with not much explanation other than, the effective rank of the attention outputs at layer 26 happens to be much higher. Does this imply that setting different parameters will result in lower # of dead features in this setting?

---

> ### Author Rebuttal · Authors · 2026-03-27
>
> ### Response to Weakness 1
>
> We thank the reviewer for this observation.
>
> **Since our TopK SAE is trained with a fixed number of activated features per token (i.e., a fixed *k*), the sparsity level is explicitly controlled and remains consistent across different methods in the same experimental setting**. While ASI does not alter the enforced sparsity level, it consistently improves the sparsity-reconstruction frontier. In practice, if one aims to match the reconstruction quality of the baseline, ASI allows for a higher sparsity level by simply setting a lower *k*.
>
> ### Response to Weakness 2
>
>
> We thank the reviewer for pointing this out.
>
> We agree that **the volatility in dead feature reduction is partly due to our experimental design: when evaluating ASI across different layers, models, and datasets, we used a fixed $D_{init}$ without hyperparameter tuning**. This may be suboptimal for activations with varying distributions. However, even without tuning, ASI consistently outperforms the standard TopK baseline across all metrics (dead feature count, MSE, and delta LM loss) and achieves superior performance over the AuxK-enhanced TopK in both MSE and delta LM loss. This suggests that ASI remains highly effective and robust in practice, and we expect further gains from tuning $D_{init}$ per setting.
>
> ### Response to Question 1
>
> During training, we maintain a sampling window of $3 \times 10^7$ tokens. After each window, we identify features with an activation frequency below $10^{-7}$ as dead.
>
> After training, we run the SAE on $10^8$ tokens and similarly count features with an activation frequency below $10^{-7}$ as dead.
>
> ### Response to Question 2
>
> We thank the reviewer for this important question. We believe the batch size used for initialization can affect model quality.
>
> SVD stable: In Appendix C, we show that $10^7$ tokens yield highly stable SVD results (coefficient of variation $\approx 4.9 \times 10^{-3}$).
>
> Train stable: In practice, however, far fewer activations suffice. Using 32,768 activations, we observe that across three random seeds, the number of dead features fluctuates by fewer than 2 (absolute), and the relative variation in MSE is below $0.1\%$, indicating sufficient stability. Given that this initialization cost is negligible compared to the total training tokens (on the order of $10^9$), we did not extensively explore the lower bound of initial batch size.
>
> ### Response to Question 3
>
> In our SAE setting (detailed in Appendix E), **performing SVD** on a $[32768, 4096]$ matrix (32768 as initial batch size) takes approximately **1.22 seconds** on an H100 GPU. In comparison, the **full training** takes about **one hour**. Thus, the initialization overhead is negligible relative to the total training cost.

---

> > ### Author Rebuttal · Reviewer_iM9h · 2026-04-03
> >
> > My concerns have been resolved by the rebuttal (except perhaps the response to Weakness 2, where I recommend you can try some naive strategies to see if they help tune the important parameter that is evidently necessary). I continue to recommend acceptance.

---

### Official Review · Reviewer_DBcH · 2026-03-10

**Soundness:** 3
**Presentation:** 4
**Significance:** 2
**Originality:** 2
**Overall Recommendation:** 4
**Confidence:** 3

**Summary:**

The paper demonstrates (by experiments) that the MHSA outputs in a Transformer layer (whatever its position, the model or dataset) lives in a subspace which is much smaller than the one of the MLP parts (I guess you mean the feed-forward part ?) or its residual connections. For this, the effective rank and a "fraction of downstream-loss recovered" measured are used.
From this observation (low-rankness), the authors propose a novel initialisation of the weights of a Sparse Auto-Encoder (SAE) which aligns the weights of the encoder to the dominant singular vectors of a SVD .
The authors also demonstrates a strong correlation between the concept of low-rankness and dead features in SAE.

**Compliance With Llm Reviewing Policy:**

Affirmed.

**Key Questions For Authors:**

- part 5.2: how can the optimal D_{init} be found ?
- does the low-rankness of MHSA evolves over layers (depth) ?
- line 151 right column: what is exactly (how is computed) this auxiliary loss (is it also the one used in AuxK in Figure 7 ?)

**Limitations:**

yes

**Strengths And Weaknesses:**

Strengths:
- the paper is very pedagogical, (mostly) clearly explained with a nice usage of colours matching figures lines to text
- the finding are clearly (experimentally) and in a convincing manner demonstrated
- the Active Subspace Initialisation method is novel, interesting and seems powerful for a better usage of the feature space and in terms of downstream losses

Weaknesses
- the paper would benefit by indicating which parts rely on SAE and which do not (since the concept of dead features only apply to the later)
- equations are not numbered,
- line 131 right column: W_h^O should be W_i^O
- line 190 left column: the use of A is confusing since A_i was the attention weights before and is now define as a place-holder for whatever (O, Residual or MLP)
- line 187 right column: "we compute the fraction of .... under full-activation" this whole paragraph could be clarified
- Figure 4 dot not seem to indicate normalized values as mentioned in the y-label
- Figure 8: which value of D_{init} (or in terms of percent) has been used for the figures ?

---

> ### Author Rebuttal · Authors · 2026-03-27
>
> ### Response to Weakness 1
> We sincerely thank the reviewer for this valuable suggestion, which will improve readability for the non-SAE community. In our paper, **Section 4 discusses attention low-rankness without relying on SAE-specific concepts, while Section 5 covers training SAEs on low-rank activations, which is SAE-related.**
>
> ### Response to Weakness 2 3 4
> We appreciate the reviewer’s careful reading and will address these issues in the camera-ready version.
> ### Response to Weakness 5
> We apologize for the confusing description of this metric. In Appendix B, we provide a formal definition. **The metric measures how many singular components need to be used without significantly affecting downstream tasks.** Results confirm that attention outputs require very few components. We will clarify this description in the main text for the camera-ready version.
>
> ### Response to Weakness 6
> We apologize for the lack of clarity. **Each line in the figure is normalized by its mean rather than its maximum**. We will update the figure labels and caption accordingly. Nevertheless, **this does not affect our conclusion**.
>
> ### Response to Question 1
> Aside from grid search, we currently cannot derive the optimal $D_{init}$ directly from activation metrics. However, as shown in Figure 6 and Appendix J.3, **there is a large range of suboptimal values**. For a $d_{model}=4096$ model, choosing $d_{init}$ from 256 to 2048 yields similar near-optimal performance. Thus, in practice, a coarse search over one or two values (e.g., $d_{model}/2$ and $d_{model}/4$) suffices to achieve near-optimal results.
>
> ### Response to Question 2
> As reported in Appendix D.2 and Figure 11, effective rank varies across layers: **early and late layers have slightly lower rank, while middle layers are slightly higher**. Nonetheless, the overall trend of low-rank attention outputs persists across layers.
>
> ### Response to Question 3
> This method is proposed by Gao et al. [1]. Specifically, they define the auxiliary loss $L_{aux} = |e-\hat{e}|_2^2$,
>
> where $e$ is the reconstruction error and $\hat{e}$ is the reconstruction using the top-$k_{aux}$ dead latents.
> Further details can be found in Appendix A.2 of [1]. It is exactly the AuxK in Figure 7.
>
> **Reference:**
>
> [1] Gao et al., 2024: Scaling and Evaluating Sparse Autoencoders.

---

> > ### Author Rebuttal · Reviewer_DBcH · 2026-04-03
> >
> > My concerns and questions have been resolved by the rebuttal.

---

### Official Review · Reviewer_7dzh · 2026-03-11

**Soundness:** 4
**Presentation:** 4
**Significance:** 2
**Originality:** 2
**Overall Recommendation:** 4
**Confidence:** 3

**Summary:**

This paper studies the low-rankedness which naturally occurs in the attention output layer of transformers, and introduces a drop-in initialization method to improve sparse auto-encoders trained on such models.

**Compliance With Llm Reviewing Policy:**

Affirmed.

**Key Questions For Authors:**

My knee-jerk reaction to the paper is the following: why do the authors not try to fix this low-rank issue? While providing an initialization method for SAE is a nice contribution, providing a solution to the dead unit problem would be a significantly more impactful contribution.
Intuitively, since the authors explain that the low-rankedness comes in great part from the overlap among the subspaces of the heads of the output projection, one would think: can't we initialize (or constrain via some penalty) the heads so that their subspaces overlap minimally? Have the authors tried this?

**Limitations:**

Adequately addressed

**Strengths And Weaknesses:**

Overall, I enjoyed reading this paper: it is well-written, easy to follow, and presents interesting and (as far as I know) new results.

Strengths:
- Soundness: all the analyses presented are sound, and the theoretical parts are well formalized.
- Presentation: the paper is particularly clear and well written. In particular, the authors make an effort to avoid excessive technicality and provide some intuition, e.g. "an intuitive explanation is that although each attention head contributes a d head -dimensional subspace, the superposition of attention heads (Jermyn et al., 2024; He et al., 2025) inevitably induces overlaps among these subspaces."
- Significance: understand precisely what causes dead units and improving SAE are valuable contributions
- I like that the authors are very up-front about limitations: they explicitly show failure modes of their initialization method

Weaknesses:
- Significance: as acknowledged by the authors, the practical usefulness of these results is limited, but this is not a weakness per se in my opinion
- The authors only show a correlation between low-rankedness and dead features: it feel like this link could be made causal by some additional experiments, for example by artificially reducing the rank of a given matrix (by setting some singular values to 0) and seeing how it changes the number of dead features?

---

> ### Author Rebuttal · Authors · 2026-03-27
>
> ### Response to Weakness 1
>
> We appreciate your clarification. We agree that **such foundational insights are often a prerequisite for downstream practical impact, especially in areas related to understanding and improving model behavior.**
>
> ### Response to Weakness 2
>
> We sincerely thank the reviewer for this valuable suggestion. Following your proposal, **we conducted additional experiments to further examine the causality between activation rank and dead features**. Specifically, we artificially modulated the effective rank of activations and trained SAEs on them, then measured the resulting proportion of dead features:
>
> | Fraction of Effective Rank (%) | Dead Features (%) |
> |--------------------------------|-------------------|
> | 92.21                          | 0.05              |
> | 90.13                          | 1.49              |
> | 87.12                          | 5.30              |
> | 83.22                          | 11.02             |
> | 78.80                          | 20.37             |
> | 73.76                          | 30.89             |
> | 67.98                          | 39.72             |
> | 61.19                          | 45.00             |
>
> **These results provide clear evidence of a causal relationship**. We will include these results in the camera-ready version.
>
> ### Response to Question 1
> We appreciate your insightful and forward-looking question. We agree that directly addressing low-rankness could lead to a more impactful contribution. In fact, we have conducted several preliminary explorations on mitigating low-rankness, though with very limited success:
>
> 1. Adjusting attention hyperparameters (inconclusive).
> Gemma-2-9B is configured with $n_{head} \times d_{head} = 4096 > d_{model} = 3584$, compared to other models with $n_{head} \times d_{head} = d_{model}$ (Llama-3.1-8B, Qwen-3-8B). Gemma-2-9B exhibits slightly higher effective rank (~62% vs ~60%), **suggesting a potential, though limited, impact of hyperparameter scaling**.
>
> 2. Architectural modifications (potentially promising).
> Gated attention [1] introduces nonlinearity that may mitigate low-rankness. To test this, we conducted experiments on the public models [2] from [1]. With other settings held constant, **gated attention shows modest improvement over standard attention** (63% vs 59%).
> Additionally, **models that do not use Grouped Query Attention** (e.g., GPT-2, Pythia) **exhibit higher effective rank compared to GQA models** (Llama-3.1, Qwen-3) (~68% vs ~60%).
>
> We note that any hyperparameter or architectural adjustments must carefully consider efficiency trade-offs.
>
> 3. Auxiliary loss during pretraining (unsuccessful).
> We add an **auxiliary loss** to increase attention rank. **Although it raises rank, it does not improve ce-loss and significantly increases training time.**
>
> 4. Gradient analysis.
> We find directions with near-zero singular values often have the largest gradient norms, suggesting **low-rankness is not due to lack of optimization signal** and helping explain why rank-targeting auxiliary losses fail.
>
> Overall, our current evidence suggests that low-rankness is likely an inherent limitation of standard attention mechanisms, and alleviating it may improve expressiveness. We leave this as an important direction for future work. We believe this strengthens, rather than detracts from, the contributions of our paper.
>
> **References:**
>
> [1] Qiu et al., 2025: Gated Attention for Large Language Models: Non-linearity, Sparsity, and Attention-Sink-Free.
>
> [2] https://huggingface.co/QwQZh/gated_attention

---

> > ### Author Rebuttal · Reviewer_7dzh · 2026-04-03
> >
> > Thanks for the rebuttal, and for including these additional results! Regarding the response to question 1: I think it is good practice to document such negative results, could you perhaps add them to supplementary material?

---

> > > ### Author Response · Authors · 2026-04-03
> > >
> > > We sincerely thank you for the helpful suggestion and for your positive feedback on our additional analysis.
> > >
> > > We fully agree that documenting such negative results would be valuable to the community. We would be glad to include these findings as supplementary material to improve the completeness and transparency of our work. However, due to ICML rebuttal constraints, we are unable to modify the PDF or upload additional files at this stage. We will incorporate these results into the paper in the camera-ready version.
> > >
> > > Thank you again!

---

### Official Review · Reviewer_TZni · 2026-03-13

**Soundness:** 3
**Presentation:** 3
**Significance:** 2
**Originality:** 2
**Overall Recommendation:** 3
**Confidence:** 3

**Summary:**

The paper studies the effective dimensionality of outputs of Transformer blocks including self-attention and MLP layers. Authors find out that across many LLM families, multi-head self-attention outputs are consistently lower-rank (sometimes as low as 60% of available dimensions) compared to MLP layers. The MHSA output projection matrix is shown to be a cause of this phenomenon. Building on this observation, activa subspace initialization (ASI) is proposed to initialize SAEs within the active subspace covered by the output. A related phenomenon, so-called Dead Features, is observed to decrease from 87% to about 1% using ASI in some cases.

**Compliance With Llm Reviewing Policy:**

Affirmed.

**Final Justification:**

**Final recommendation:** 3 (but I am not strictly against acceptance)

In my last reply, I asked the authors to compare the output rank against $\sum_j \text{rank}(H_j)$ instead of $d_{model}$. Since [1] establishes that individual head outputs $H_j$ are low-rank, the true theoretical upper bound is: $\text{rank}(\text{MHA}) \le \sum_{j=1}^{h} \text{rank}(H_j)$. Consequently, if $\text{rank}(\text{MHA})$ is simply comparable to $\sum_j \text{rank}(H_j)$, the authors' finding is a corollary of [1]. Since I am uncertain if the central claim of the paper is substantially novel, my final recommendation is 3 (but I am not strictly against acceptance)

**Key Questions For Authors:**

See Weaknesses and Questions above.

**Limitations:**

yes

**Strengths And Weaknesses:**

**Strength:**
- The claimed phenomenon is observed across multiple model families to support its generality
- Some of the mechanistic reasons behind the observation are studied
- The proposed method, ASI, is computationally inexpensive way of significantly improving SAE training.

**Weaknesses and Questions:**
- Low-rankness of attention outputs was also observed in [1], attributed similarly to the projection $W_V W_O$ (Section 4.1 of [1]). Then, [1] proposed to use nonlinear activation within the self-attention layer to mitigate low-rankness to enhance expressivity. Can authors compare their findings versus [1]? Would ASI be still beneficial in attention layers of type [1]? This is important since Gated Attention layers have been shown quite effective in LLMs. This might have impact on novelty and significance of the current submission.
- Authors define a "dead feature" as a feature that does not activate for 10 million tokens. While this is based on a previous work as cited, it still sounds somewhat arbitrary. It would be better to have some analysis/discussion to differentiate between truly dead features and "rare but important" features. This requires rigorous definitions with clear motivations behind them.
- If the attention outputs have about 60% effective rank, then the MLP layer increases it to 90%, what is the mechanism behind rank recovery? How does ASI interact with this rank phase between attention and MLP layers?
- Does the low-rankness observation hold for quantized (or even be more pronounced) so that ASI is more effective in that setting?
- Since authors figure $W^O$ to be a major cause of low-rankness in MHSA output, could they suggest a way to alleviate this? Is low-rankness of MHSA output "a bug or a feature"?

___

**References:**

[1]: Qiu, Z., Wang, Z., Zheng, B., Huang, Z., Wen, K., Yang, S., Men, R., Yu, L., Huang, F., Huang, S., Liu, D., Zhou, J., & Lin, J. (2025). Gated attention for large language models: Non-linearity, sparsity, and attention-sink-free. In Proceedings of the Thirty-Ninth Annual Conference on Neural Information Processing Systems (NeurIPS 2025).

---

> ### Author Rebuttal · Authors · 2026-03-27
>
> ### Response to Weakness 1
> We apologize for not clearly distinguishing our work from [1]. While both study low-rankness in attention, **[1] focuses on single-head properties**, whose $W_V$ and $W_O$ form a low-rank linear mapping. In contrast, **we identify a distinct phenomenon: the final output of the entire attention block is low-rank**, which, to our knowledge, has not been previously reported.
>
> Gated attention introduces nonlinearity that may mitigate low-rankness. To test this, we run additional experiments on three public models [2] from [1]:
>
> | attention type | model name| attention output|MLP output|residual stream|
> |-|-|-|-|-|
> | standard |1B_baseline| 1207|**1763**|1683|
> | gated|1B_gate_headwise|1196|1758|1680|
> | gated|1B_gate_elementwise|**1279**|1754|**1684**|
>
> Compared to standard attention, **although 1B_gate_elementwise slightly improves, gated attention still exhibits low-rank attention outputs.**
>
> We will include these results in the camera-ready to further highlight our novelty and significance.
>
> ### Response to Weakness 2
> We adopt $10^7$ tokens as the dead feature threshold for two reasons:
>
> (1) Bricken et al. [3] shows **almost no interpretable features in the ultralow cluster (freq ~ $10^{-7}$)**. Our manual inspection of “dead features” in our SAEs also finds them hard to interpret. These suggest that they are not “rare but important.”
>
> (2) Prior work from Anthropic [4] and OpenAI [5] uses the same threshold ($10^7$), enabling fair comparison with their auxiliary loss methods.
>
> ### Response to Weakness 3
>
> In a transformer with residual connections, layer i performs:
>
> $resid_{pre}^i = resid_{post}^{i-1} $
>
> $resid_{mid}^i = resid_{pre}^i + attn_{out}^i $
>
> $resid_{post}^i = resid_{mid}^i + MLP_{out}^i $
>
> We find that, $resid_{pre}$, $resid_{mid}$, $resid_{post}$, and $MLP_{out}$ have ~90% effective rank, while $attn_{out}$ has ~60%. Hence, **the high rank of $resid_{mid}$ is mainly preserved by $resid_{pre}$ via the residual path**.
>
> (We may have misunderstood your question and would appreciate clarification if needed.)
>
> ASI is only used when training SAEs on activations (e.g., $attn_{out}$) to reduce dead features, and **does not modify model computations or affect rank dynamics**.
>
> ### Response to Weakness 4
>
> **We evaluate the effective rank on both the original Qwen3-8B and its quantized variant Qwen3-8B-FP8**. The results show nearly identical mean effective ranks across layers:
>
> | Component  | Qwen3-8B| Qwen3-8B-FP8 |
> |-|-|-|
> |attention|2368|**2373**|
> |MLP|**3486**|3484|
> |residual| 3390|**3395**|
>
> These findings indicate that the low-rankness of attention outputs persists after quantization. Although the exact behavior may vary with different quantization schemes, **the discrepancy is minimal** in this case.
>
> ### Response to Weakness 5
>
> From a variance decomposition view, $W^O$ limits attention output variance along certain directions. From backpropagation, its compression of $Z$ suggests these directions may carry redundant or less useful information, implying **the issue lies in the attention mechanism as a whole rather than $W^O$ alone.**
>
> Regarding alleviating low-rankness, we conduct several preliminary explorations:
>
> 1. **Adjusting attention hyperparameters** (inconclusive).
> Gemma-2-9B is configured with $n_{head} \times d_{head} > d_{model} $, compared to other models with $n_{head} \times d_{head} = d_{model}$ (Llama-3.1-8B, Qwen-3-8B). Gemma-2-9B exhibits slightly higher effective rank (~62% vs ~60%).
>
> 2. **Architectural modifications** (potentially promising).
> Gated attention slighly increases rank (1207 → 1279).
> Additionally, **models that do not use GQA** (e.g., GPT-2, Pythia) **exhibit higher effective rank compared to GQA models** (Llama-3.1, Qwen-3) (~68% vs ~60%).
>
> 3. **Auxiliary loss during pretraining** (unsuccessful).
> We add an auxiliary loss to increase attention rank. Although it raises rank, it does not improve final loss and significantly increases training time.
>
> 4. **Gradient analysis**.
> We find directions with near-zero singular values often have the largest gradient norms, suggesting low-rankness is not due to lack of optimization signal and helping explain why rank-targeting auxiliary losses fail.
>
> Overall, **current evidence suggests low-rankness in standard MHSA is more a limitation**. However, we do not yet have a definitive solution and leave this for future work.
>
> **References:**
>
> [1] Qiu et al., 2025: Gated Attention for Large Language Models: Non-linearity, Sparsity, and Attention-Sink-Free.
>
> [2] https://huggingface.co/QwQZh/gated_attention
>
> [3] Bricken et al., 2023: Towards monosemanticity: Decomposing language models with dictionary learning.
>
> [4] Templeton et al., 2024: Scaling Monosemanticity: Extracting Interpretable Features from Claude 3 Sonnet.
>
> [5] Gao et al., 2024: Scaling and Evaluating Sparse Autoencoders.

---

> > ### Author Rebuttal · Reviewer_TZni · 2026-04-02
> >
> > I thank the authors for extensive rebuttal and new results.
> >
> > Could the authors clarify if the aggregate (multi-head) low-rankness they observed is significantly lower than the theoretical maximum rank allowed by the sum of individual low-rank head projections (since $rank(H_1+H_2) \le rank(H_1) + rank(H_2))$? Without this comparison it is hard to tell if multi-head low-rankness is truly a novel phenomenon to find or an implication of [1].

---

> > > ### Author Response · Authors · 2026-04-02
> > >
> > > We are glad that the discussion has helped address part of your concern, and we appreciate this important follow-up question.
> > >
> > > Our key finding is that **the aggregate (multi-head) rank is significantly lower than the theoretical maximum rank implied by summing the individual low-rank head projections**.
> > >
> > > More specifically, [1] states that “we can merge $W_V^k W_O^k$ into one low-rank linear mapping applied over all $X_j$ as $d_k < d_{model}$”, where $d_k$ corresponds to $d_{head}$ in our paper. They characterize the rank of the projection for each individual head, indicating that the output of a single head is at most rank $d_{head}$. Following [1], this implies that the theoretical maximum rank allowed by the additive bound $rank(H_1 + \cdots + H_k) \le \sum_j rank(H_j)$ is upper bounded by $n_{heads} \cdot d_{head}$.
> > >
> > > For nearly all models we study, this quantity satisfies $n_{heads} \cdot d_{head} = d_{model}$, which corresponds to the standard design of multi-head attention. (The only exception in our experiments is Gemma-2-9B, where $n_{heads} \cdot d_{head} = 4096 > d_{model} = 3584$.)
> > >
> > > Therefore, based on [1], the implied upper bound for the aggregate multi-head rank is $d_{model}$. However, our paper shows that the observed aggregate rank is only around $0.6 * d_{model}$, which is substantially below this theoretical maximum.
> > >
> > > We summarize the corresponding statistics below:
> > >
> > > | Metric | GPT-2 | Llama-3.1-8B | Gemma-2-9B | Qwen3-8B |
> > > |---|---:|---:|---:|---:|
> > > | $n_{head}$ | 12 | 32 | 32 | 32 |
> > > | $d_{head}$ | 64 | 128 | 128 | 128 |
> > > | $n_{head} \cdot d_{head}$ | 768 | 4096 | 4096 | 4096 |
> > > | $d_{model}$ | 768 | 4096 | 3584 | 4096 |
> > > | rank of multi-head attention output | 514 | 2506 | 2232 | 2376 |
> > >
> > > These results demonstrate that our finding is not merely inherited from the single-head analysis of [1], but reflects a novel and nontrivial phenomenon at the multi-head level.

---

### Decision · Program_Chairs · 2026-04-30

**Decision:**

Accept (regular)

**Comment:**

This paper received three positive reviews and one negative one. All reviewers agree that understanding effective dimensionality of attention outputs in LLMs is interesting, that the empirical observations are consistent across multiple model families, and that ASI is a practically useful and computationally inexpensive improvement to SAE training. The main concerns are about the novelty of the low-rankness observation relative to prior work, and the somewhat indirect link between low-rankness and downstream performance impact.

The main argument toward rejection from reviewer **TZni** is that the rank of the multi-head attention output should be compared to the sum o the rank of each head, instead of the dimension, as results from [A] already shows that each head is low-rank. The comparison provided in the rebutal uses a very loose lower bound rather than what is required by the reviewer, and their question remains unresolved. This needs to be addressed in the manuscript and the contribution/link to [A] need to be revised accordingly.

A second concern shared by reviewers **TZni**, **7dzh**, and **iM9h** is whether low-rankness is a feature, or a bug that can be fixed, and how dead features relate to downstream performance. The rebuttal added a rank-manipulation experiment that convincingly established causality between rank and dead features. That is kind of expected from dictionary learning literature, where it is well established that dictionary elements are only learned in the span of the element from which they are derived (see [B, Proposition II.1] for instance). The link to downstream performance is shown via the delta LM loss metric, though this remains an indirect measure. Reviewer **7dzh** also asked whether one could directly fix the low-rankness rather than working around it: the authors documented several failed attempts (hyperparameter tuning, auxiliary loss during pretraining, architectural modifications), which are negative results worth preserving.

Overall, the paper makes a good empirical contribution with a practical and well-motivated application. I recommend acceptance, and encourage the authors to address the following points in the final version:

- Compare the aggregate multi-head attention output rank explicitly against the tighter bound $\sum_h \text{rank}(W_O^h)$ to establish novelty relative to [A], or revise the framing of the title and abstract to emphasize the SAE application rather than the low-rankness observation as a standalone finding.
- Include the negative results on alleviating low-rankness (failed auxiliary losses, architectural variants) in the supplementary material, as suggested by reviewer **7dzh**.
- Show stability of the effective rank computation with respect to the number of tokens used (e.g., by varying token count and reporting variance), to support the choice of 10M tokens to compute it.
- Discuss the volatility of dead feature reduction across layers (Table 7, layer 26) and tune $D_\text{init}$ per setting, as flagged by reviewer **iM9h**.

[A] Qiu et al. *"Gated Attention for Large Language Models: Non-linearity, Sparsity, and Attention-Sink-Free."* NeurIPS 2025.
[B] Malézieux et al. *"Where prior learning can and can't work in unsupervised inverse problems"* 2024.